# Alleviating Overgrazing Stress and Promoting Grassland Plant Regeneration via Root Exudate-Mediated Recruitment of Beneficial Bacteria

**DOI:** 10.3390/microorganisms13061225

**Published:** 2025-05-27

**Authors:** Ting Yuan, Jiatao Zhang, Shaohong Zhang, Shuang Liang, Changhong Zhu, Weibo Ren, Jialu Liang

**Affiliations:** 1Inner Mongolia Key Laboratory of Grassland Ecology, and the Candidate State Key Laboratory of Ministry of Science and Technology, Inner Mongolia University, Hohhot 010010, China; yuanting9541@163.com (T.Y.); jtzhang0209@163.com (J.Z.); 13453024677@163.com (S.Z.); 13847802006@163.com (S.L.); 18147178516@163.com (J.L.); 2National Center of Pratacultural Technology Innovation, Hohhot 010070, China; yuanting9541@139.com

**Keywords:** overgrazing, *Leymus chinensis*, plant growth-promoting bacteria (PGPR), *Paraburkholderia graminis*, root exudate, multi-omics analysis

## Abstract

Overgrazing (OG) is an important driver of grassland ecosystem degradation and productivity decline. Plants may effectively cope with OG stress by regulating their synergistic interactions with plant growth-promoting rhizobacteria (PGPR) through root exudates. However, the synergistic regulatory mechanisms remain unclear. Under OG stress, *Leymus chinensis* recruited the specific PGPR strain *Paraburkholderia graminis* (B24) by regulating specific root exudate compounds, including amino acids, alkaloids, and organic acids, which enhance B24 chemotaxis and biofilm formation. The B24 inoculation systematically regulated the transcription of key plant growth and development genes, including those involved in nutrient transport and cell wall expansion, which enhanced nutrient uptake and promoted the overall growth of *L. chinensis*. Furthermore, B24 regulated the homeostasis of endogenous *L. chinensis* through the synergistic effects of hormones and the trade-off between growth and defense. Integrated transcriptomic and metabolomic analyses revealed that B24 regulation enhanced carbon and nitrogen metabolism, and energy supply after mowing, forming a holistic adaptive mechanism that enabled *L. chinensis* to effectively recover from mowing-induced stress, thereby improving its adaptability and regenerative capacity. This study provides a scientific basis and support for elucidating the response mechanisms of how grassland plants cope with OG stress, optimizing grassland management, and rapidly restoring and enhancing grassland productivity.

## 1. Introduction

Grasslands play a key role in maintaining global ecosystem functions and are widely utilized for livestock production [1]. However, climate change and improper anthropogenic disturbances—particularly livestock overgrazing—have led to severe grassland degradation in many regions [2]. In recent years, the response mechanisms of grassland plants to overgrazing stress have become a hotspot in ecological research. Additionally, trampling during grazing may affect soil moisture, aeration, and redox treatments, while animal feces and urine can enhance soil nutrient availability and increase the decomposable organic carbon content, thereby shaping microbial community composition and function [3,4]. Additionally, overgrazing reduces aboveground vegetation biomass, triggering plant defense responses that redirect biomass allocation to roots and increase the release of specific root exudates, thereby impacting soil and microbial communities [5,6]. However, the effects of plant interactions with rhizosphere microbial under overgrazing stress remain unknown. Mowing, another common grassland management practice worldwide, acts as a critical driver of nutrient cycling in grassland ecosystems, potentially influencing ecosystem functions [7]. Similar to grazing, moderate mowing can promote compensatory plant growth, enhancing photosynthetic efficiency and grassland productivity [8]. Numerous studies have demonstrated that short-term mowing enhances nutrient cycling, facilitates plant nutrient acquisition, and enhances community productivity [9,10]. The reallocation of nutrients from roots to shoots is a crucial strategy by which grassland plants adapt to grazing or mowing through shoot regeneration [11].

Plant–microbe interactions are crucial for promoting plant growth and enhancing stress tolerance. With advancements in omics technologies, the synergistic regulatory mechanisms of plant–microbe interactions have been increasingly elucidated. Although the detailed mechanisms by which plant growth-promoting rhizobacteria (PGPR) regulate plant growth are not fully understood, certain well-characterized PGPR strains are widely used in agriculture and grassland ecosystem to stimulate growth and increase yield in plants [12]. Recent studies have shown that certain PGPR strains, such as *Paraburkholderia* could influence host plant nutrition absorption and transportation, growth and development, and stress resistance. *Paraburkholderia* may serve as a specific component of core *Poaceae* microbiota and play a vital role in promoting plant growth [13]. Previous studies have also shown that *Paraburkholderia*/*Burkholderia* is a beneficial class of endophytes for improving the growth of sugarcane, maize, wheat, and rice [14]. There are also studies that indicate that beneficial microbes regulate the genes involved in secondary metabolite biosynthesis, carbohydrate metabolism, and nutrient and sugar absorption, which crucial are for fostering plant growth and development [15]. In addition to enhancing hormone synthesis and nutrient availability, PGPR can improve plant regeneration capacity and stress tolerance post-herbivory [16].

Plants employ a range of strategies to respond to biotic and abiotic environmental factors, including the release of root-derived compounds into the surrounding soil. Root exudates play a crucial role in plant–soil–microbe interactions, altering the physicochemical environment of the rhizosphere soil, facilitating nutrient acquisition, and recruiting specific beneficial microbes to optimize plant growth and performance [17]. Primary metabolites such as carbohydrates, amino acids, and organic acids, released by roots, provide energy and nutrients for soil microbes, while secondary metabolites such as flavonoids, terpenoids, and phenolics act as signaling molecules, attracting specific microbes to the roots [18]. Under biotic or abiotic stress, plants can alter their root exudation patterns to selectively recruit beneficial rhizosphere microbes, enhancing stress resistance. PGPR utilize chemotaxis to sense root exudate signals, moving towards and colonizing plant roots via flagella, eventually forming biofilms on root surfaces. It has been reported that the local infection of cucumber by *Fusarium oxysporum* increases the tryptophan content and decreases the raffinose content, which is related to beneficial *Bacillus amyloliquefaciens* colonization. Similarly, legumes secrete higher levels of flavonoids to attract nitrogen-fixing bacteria under nitrogen-deficient treatments [19]. Additionally, maize roots secrete benzoxazinoids as secondary metabolites to attract microbes, enhancing plant defense [20]. Studies have indicated that under herbivory stress from insects, plants can actively regulate root exudates to shape rhizosphere microbial communities, which can trigger immune responses in aboveground organs, enhancing resistance to herbivores [21]. Furthermore, PGPR can stimulate plant growth by promoting hormone or nutrient production and can trigger immune responses in above- or below-ground tissues, improving the plant regeneration capacity or stress tolerance post-herbivory [16].

Overgrazing, as an important biological stress, may also trigger a similar recruitment effect. Plants may attract beneficial microorganisms (such as phosphorolytic bacteria, *Azotobacter,* and PGPR) to colonize the rhizosphere by secreting specific amino acids, sugars, organic acids, and secondary metabolites to help plants better adapt to the stressed environment, thus improving their tolerance to overgrazing stress [17]. These recruited beneficial microbes may promote nutrient acquisition, produce plant hormones, or induce stress resistance, potentially exhibiting stronger ecological adaptability under grazing stress. Previous studies have demonstrated that long-term overgrazing significantly alters the rhizosphere microbial communities in grassland plants, with these changes having certain growth-promoting effects on plant growth [22]. Additionally, our earlier research has confirmed that grassland plants can reshape beneficial microbial communities by altering root metabolite composition, which helps plants to resist grazing stress and enhances environmental adaptability [23,24]. These studies have provided crucial insights into the plant–microbe synergistic effects under overgrazing stress. However, the dynamics of plant–PGPR interactions in response to large herbivore grazing stress remain poorly understood, with a critical knowledge gap in the underlying coordination mechanisms and regulatory pathways that govern these complex interactions.

This study focused on *Leymus chinensis*, a dominant species in Inner Mongolia’s grasslands, and employed a multi-omics approach (high-throughput sequencing, transcriptomics, and metabolomics) to identify core PGPR, screen them for beneficial strains, identify key root exudate components, analyze the interaction effects, and finally to clarify the potential mechanisms by which the specific PGPR strain *Paraburkholderia graminis* (B24) promotes plant regeneration. Our findings provide a scientific basis and support for understanding grassland plant responses to grazing stress, optimizing grassland management, and rapidly restoring and enhancing grassland productivity.

## 2. Materials and Methods

### 2.1. Study Site and Sampling

The study area was located at the Inner Mongolia Grassland Ecosystem Research Station (43°38′30″ N, 116°42′20″ E) in xilinhot city of northern China. This region experiences a temperate semi-arid climate [25], with an average annual precipitation of 388.2 mm from 2012 to 2022, of which 83.2% occurs between May and September. The study site is separated into two areas by a pasture fence: (1) a no grazing (NG) area (600 m × 400 m; in this enclosed grazing-exclusion area, long-term ecological observation studies have been conducted since 1983) and (2) an overgrazed (OG) area (600 m × 100 m; this area has been subjected to overgrazing [~3 sheep/ha] for >50 years from June to October every year) [26]. The NG and OG areas were continuously distributed over the same upper basalt platform, helping to control soil heterogeneity. During the peak plant growth period in July 2022, rhizosphere soil of *L. chinensis* was collected from both the NG and OG plots, following the method described by Chen et al. (2020) [27]. In the NG and OG plots, three transects were established approximately 30 m apart. Along each transect, five rhizosphere soil samples were collected and combined into a composite rhizosphere soil sample, resulting in a total of three replicate rhizosphere soil samples.

*Leymus chinensis* root exudates were collected using a modified method based on Ren et al. (2008) [28] and Luo et al. (2020) [29]. *L. chinensis* plants were collected from NG and OG plots, ensuring that the roots remained as intact as possible. After cleaning the roots with sterile deionized water, 20 *L. chinensis* plants with similar growth trends were placed in beakers containing Hoagland’s complete nutrient solution, and their roots were submerged for 3 days for regeneration. The beakers were wrapped in tinfoil to protect them from light and kept in a growth chamber at 28 °C with aeration for 6 h/day to repair any root damage. After regeneration, the roots were transferred to beakers containing 1000 mL sterile distilled water and incubated at 28–30 °C for 12 h. The collected root exudates were then filtered, concentrated to dry matter under reduced pressure, and stored at −80 °C for metabolomic profiling [30].

### 2.2. Amplicon Sequencing of 16S rRNA Gene of L. chinensis Rhizosphere Soil

Amplicon sequencing was performed to determine whether B24 can successfully colonize and alter the rhizosphere microbial community. A HiPure Stool DNA Kit (Magen, Guangzhou, China) was used to obtain the total DNA from *L. chinensis* rhizosphere soil. The V3–V4 region of the 16S rRNA gene was amplified using the specific primers 341F (5′-CCTACGGGNGGCWGCAG-3′) and 806R (5′-GGACTACHVGGGTATCTAAT-3′) [31]. The purified PCR products were sequenced on an Illumina NovaSeq 6000 SP PE250 platform by Gene Denovo Biotechnology Co., Ltd. (Guangzhou, China). Raw reads were assembled and quality-filtered using FASTP v0.18.0. Representative amplicon sequence variants (ASVs) were classified using RDP classifier v2.2 [32]. Abundance statistics for each species classification were displayed using Krona (version 2.6). Intergroup shared or endemic species were plotted using Venn diagrams using the VennDiagram package (verison 2.12.1) and biomarker species in each group were screened using LEfSe software (version 1.0).

### 2.3. Bacterial Isolation, Identification, and Testing of Growth-Promoting Properties

*Paraburkholderia graminis* was significantly enriched following overgrazing and identified as a potential biomarker, suggesting its key role in mitigating grazing stress. Thus, we isolated and purified the strain for further investigation. Culturable bacteria were isolated from *L. chinensis* rhizosphere soils of the OG treatments using the plate culturing method as previously described [33]. The 16S rRNA gene of strains were amplified with primers F27 (59-AGTTTGATCMTGGCTCAG-39)/R1492 (59-GGTTACCTTGTTACGACTT-39). After the elimination of potential clonal duplicates, we obtained 15 Parburkholderia strains. The sequencing results were analyzed by NCBI BLAST (https://blast.ncbi.nlm.nih.gov/Blast.cgi, accessed on 1 April 2025), and the phylogenetic tree was established by MEGA 11 software. From these strains, we selected B24 to study the growth-promoting mechanism of *L. chinensis*, which displayed 100% clustering similarity with Parburkholderia graminis (Appendix A). The growth-promoting properties (phosphate-solubilizing capacity, IAA production, nitrogenase activity, and ACC deaminase activity) of B24 was assessed as described by Yuan et al. (2025) [34].

### 2.4. Metabolomic Profiling of L. chinensis Roots and Root Exudates

Metabolomic analyses were conducted as previously described [35] on *L. chinensis* roots after B24 inoculation, as well as on root exudates collected from plants under NG and OG treatments. Briefly, *L. chinensis* root samples (100 mg) were ground to a fine powder using a mixer mill (Retsch GmbH, Düsseldorf, Germany). Metabolites were extracted with 80% aqueous methanol, centrifuged, filtered (0.22 m pore size) for an ultra-performance liquid chromatography-tandem mass spectrometry (UPLC-MS/MS) analysis conducted by Gene Denovo Biotechnology Co., Ltd. (Guangzhou, China). Additionally, the collected root exudates (10 mL) from the NG and OG groups (in Section 2.1) were lyophilized, dissolved in 80% methanol, and analyzed using the same procedure. Differential expressed metabolites (DEMs) were identified based on a threshold of *p* < 0.05 and variable importance in the projection (VIP) > 1.

### 2.5. In Vitro Chemotaxis and Biofilm Formation Assays

The effects of the specific root exudate compounds—2-phenylglycine (CAS: [2835/6/5], Macklin), L-homocysteine (CAS: [626-72-2], Macklin), D-asparagine (CAS: [2058-58-4], Macklin), L-leucyl-L-alanine (CAS: [7298-84-2], Macklin), β-1-Tomatine (CAS: [17406-46-1], Macklin), and α-keto-γ-(methylthio)butyric acid (CAS: [51828-97-8], Macklin)—on B24 chemotaxis were measured using modified capillary assays based on the protocol by Rudrappa et al. (2008) [36] and Gordillo et al. (2007) [37]. Briefly, a 25-gauge needle was attached to a 1 mL syringe containing 100 μL different concentrations of root exudate compounds or phosphate buffer solution (negative control) and the needle of the syringe was inserted into the B24 bacterial suspension. After static incubation for 2 h at 30 °C, the solution in the syringe was diluted, and the number of B24 cells was counted on Luria–Bertani (LB) agar. The effects of the root exudate compounds on B24 biofilm formation were measured using biofilm formation [38,39]. The B24 bacterial suspension (OD_600_ = 1) was centrifuged, washed, and resuspended in 1/2 MSgg minimal salts, glutamate, and glycerol (MSgg) medium. Each well of a 48-well microtiter plate was inoculated with freshly prepared 1/2 MSgg medium and B24 suspension. Next, 10 μL root exudate compound or 1/2 MSgg medium (negative control) was added to each well. Each treatment had six replicates with four wells per replicate. After static incubation for 24 h, adhered cells were stained with 0.1% crystal violet. Biofilm formation was measured at OD_470_ using a CFX96 TM Real-Time System (Bio-Rad Laboratories, Inc., Hercules, CA, USA).

### 2.6. Pot Experiments

*Leymus chinensis* seeds (West Ujumuqin Leymus, provided by Grassland Research Institute, Chinese Academy of Agricultural Sciences, Hohhot, China) with uniform size and full shape were disinfected and germinated in petri dishes. After the seedlings reached the three-leaf stage, uniformly sized seedlings were selected and transplanted into plastic pots (20 cm high; 18 cm in diameter) containing about 3 kg of soil (mixture of field soil, nutrient soil, and vermiculite in a 4:1:1 volume ratio). Each pot was planted with eight seedlings, with four pots per treatment as replicates. The pots were randomly placed in a greenhouse (light/dark cycle of 16/8 h, day/night temperatures of 30/18 °C, relative humidity of 60–70%, and soil moisture maintained at 60%). A mowing treatment was applied to simulate grazing [34,40]. The *L. chinensis* plants were randomly divided into four groups: a control group without mowing or inoculation (BCK), a B24 inoculation treatments group (B24), a mowing treatments group without inoculation (MCK), and a mowing followed by B24 inoculation treatments group (M24). After two weeks of growth, the *L. chinensis* plants with MCK and M24 treatments were mowed (stubble height of 6 cm), while the remaining plants were allowed to grow normally. One week after mowing, 10 mL of B24 suspension (10^8^ CFU/mL) was injected into the soil around the roots of each plant in the B24 and M24 treatments groups, with a total of 80 mL inoculated per pot. Sterile water was inoculated in the same way in the control group. Four weeks after B24 inoculation, plant height, leaf width, stem diameter, leaf length, chlorophyll content, fresh shoot weight, fresh root weight, and dry shoot and root weights were measured. The fresh root samples were harvested for transcriptome and metabolome evaluation. Rhizosphere soil was collected for a rhizosphere microbiome analysis.

### 2.7. Determination of Physiological Indices

Liquid chromatography-tandem mass spectrometry (LC–MS/MS) (Rigol L3000, Beijing, China) was used to assess Abscisic acid (ABA), indole-3-acetic acid (IAA), jasmonic acid (JA), gibberellin (GA3), and salicylic acid (SA) content in L.chinensis root [41]. The glucose, sucrose, fructose glucose, total carbon (TC), total nitrogen (TN), and total phosphorus (TP) content were measured using corresponding assay kits (Suzhou Keming Biotechnology Co., Ltd., Suzhou, China) [42].

### 2.8. Transcriptome Sequencing and qRT-PCR Analysis

Transcriptome sequencing was performed on the *L. chinensis* root from the four treatment groups, with three replicates per treatment. Total RNA was isolated using the TRIzol reagent kit (Invitrogen, Carlsbad, CA, USA), and its integrity and quality were evaluated using an Agilent 2100 Bioanalyzer and NanoDrop spectrophotometer (Thermo Scientific, Waltham, MA, USA). Sequencing libraries were constructed using the NEBNext^®^ Ultra™ RNA Library Prep Kit and subjected to sequencing on the Illumina Novaseq X Plus platform at Gene Denovo Biotechnology Co. (Guangzhou, China). The expression of each transcript was assessed by per million mapped reads (TPM) using the exon model per kilobase [43], and then the expression of each gene was normalized. The DESeq2 software package was used to identify the differentially expressed genes (DEGs) [44] based on |log_2_fold change (FC)| > 1 and adjusted *p*-value (*padj*) < 0.05. Based on the transcriptomic results, several genes related to amino acid metabolism, carbohydrate metabolism, nitrate transport, phosphate transport, and sugar transport were selected for qRT-PCR validation. The primer sequences are provided in Appendix A and the relative expression levels of DEGs were calculated using the 2^−∆∆Ct^ method.

### 2.9. Statistical Analysis

The data are presented as the mean ± standard error. Means were compared by a one-way analysis of variance (ANOVA) followed by Tukey’s Studentized range (honestly significant difference [HSD]) test in IBM SPSS Statistics 26. Correlations between root exudates and rhizosphere bacteria were analyzed using the Hmsic package in R v4.0.2. Figures were created using Adobe Illustrator 2022, Origin 2022, RStudio (version 4.3.1), and the OmicShare platform (https://www.omicshare.com/ (accessed on 1 April 2025).

## 3. Results

### 3.1. Effects of Long-Term Overgrazing on the Rhizosphere Bacterial Communities

The 16S rDNA sequencing results revealed that a total of 734,764 valid sequences and 81,083 ASVs were obtained in the six rhizosphere soil samples from the OG plot. The most abundant bacterial phyla were Actinobacteria (15.11%), Proteobacteria (13.56%) Bacteroidota (12.79%), and Acidobacteriota (10.54%) (Figure 1A). The most abundant bacterial genera were RB41 (3.76%), *Candidatus_Udaeobacter* (3.53%), *Sphingomonas* (2.12%), and *Bacillus* (1.9%) (Figure 1B). Additionally, OG significantly reduced the α diversity of bacterial communities by significantly reducing the observed species (Sobs), Chao1, and abundance-based coverage estimator (ACE) indices (Figure 1C–E). Multiple analysis methods in 16S rRNA sequencing (LEfSe analysis, Venn diagrams, Welch’s *t* test) were used to identify the dominant, unique, and common microbial groups in NG and OG treatments. The LEfSe analysis revealed that 30 bacterial genera were significantly enriched after overgrazing, primarily including *Sphingopyxis*, *Lysinibacillus*, *Nitrosospira*, *Paenibacillus*, *Domibacillus*, *Solirubrobacter*, and the *Burkholderia–Caballeronia–Paraburkholderia* (BCP) group (Appendix A). Venn diagrams were constructed to highlight the unique bacterial ASVs under overgrazing, with 21,863 ASVs exclusively present in overgrazed treatments (Figure 1F), among which 82 belonged to the BCP group. Furthermore, a total of 164 bacterial species were only present in the OG treatments, among which *Paraburkholderia graminis* was relatively abundant and was the key bacterial species in the BCP group (Figure 1G). The relative abundance of this group in the OG treatments was significantly higher than that in the NG treatment (Welch’s *t*-test, *p* < 0.05) (Figure 1H). Therefore, the BCP group and *Paraburkholderia graminis*, as potential biomarkers, may play a key role in alleviating grazing stress and ensuring the grazing tolerance of *L. chinensis*.

### 3.2. Paraburkholderia graminis Isolation and Growth-Promoting Characteristics

We isolated culturable bacteria from *L. chinensis* rhizosphere soils in the OG treatments. A total of 220 bacterial isolates were isolated, among which B24 was identified as *Paraburkholderia graminis* according to the phylogenetic analysis. The dissolved inorganic phosphorus, indole-3-acetic acid (IAA) production, ACC deaminase, and nitrogenase activities of isolated *Paraburkholderia graminis* were 179.90 μg/mL, 25.33 μg/mL, 178.12 IU/L, and 199.13 IU/L, respectively (Appendix A), indicating that the strain can promote inorganic phosphate release, IAA production, ACC deaminase production, and N-fixation.

### 3.3. Identification of Key Root Exudates and the Analysis of Their Association with Specific Rhizosphere Bacteria

Through the metabolomic analysis of root exudates, a total of 144 DEMs (72 upregulated and 72 downregulated) between the NG and OG treatments were identified (Figure 2A). The enrichment of 15 DEMs primarily including amino acid and nucleotide-related DEMs after OG (Figure 2B) was significantly induced. Additionally, changes were induced in some primary and secondary DEMs after overgrazing, such as alkaloids, sugars, organic acids, and flavonoids (Figure 2C). These metabolites, as important components of root exudates, may act as chemical signaling molecules between plants and microorganisms, regulating plant–microbe interactions. By mapping these DEMs to the Kyoto Encyclopedia of Genes and Genomes (KEGG) database, the significantly enriched KEGG pathways were identified, including purine metabolism, ABC transporters, taurine and hypotaurine metabolism, nitrogen metabolism, and histidine metabolism (Figure 2D).

The significantly enriched bacterial genera BCP group, *Jatrophihabitans*, *Mesorhizobium*, *Lysinibacillus*, and *Conexibacter,* showed significant positive correlations with most root exudates after OG. These included amino acids such as L-homocystine, D-asparagine, Phosphocreatine, and Asp-glu; alkaloids such as β-1-Tomatine and Chinese bittersweet alkaloid II; flavonoids such as 5,7-Dihydroxy-3′,4′,5′-trimethoxyflavone; sugars such as 2-Deoxy-D-galactose; and terpenoids such as Lappaconitine (Figure 2E). In particular, the BCP group exhibited a highly significant positive correlation (*p* < 0.01) with the flavonoid tricin 5-O-hexoside and the organic acid alpha-keto-gamma-(methylthio)butyric acid. At the bacterial species level, the dominant species present only in the overgrazed treatments such as *Paraburkholderia graminis*, *Bacillus niacini*, *Bacillus asahii*, and *Mesorhizobium huakuii* showed significant positive correlations (*p* < 0.05) with specific root exudates, including N′-nitrosoanabasine, Asp-glu, 5,7-Dihydroxy-3′,4′,5′-trimethoxyflavone, L-homocystine, Chinese bittersweet alkaloid II, Lappaconitine, and D-asparagine (Figure 2F). Additionally, *Paraburkholderia caledonica*, *Sphingomonas mali*, *Bacillus nealsonii*, *Rhizobium mesosinicum,* and *Paraburkholderia graminis* were significantly positively correlated with alpha-keto-gamma-(methylthio)butyric acid (*p* < 0.05).

### 3.4. Effects of L. chinensis Specific Root Exudate Compounds on B24 Chemotaxis and Biofilm Formation

To explore which key differential root exudates changed under overgrazing stress, important DEMs were selected based on VIP scores (Figure 2B). Based on the above correlation analysis between root exudates and rhizosphere bacteria, we analyzed the interaction between root exudates with significantly increased relative abundance under overgrazing stress and *Paraburkholderia graminis* (B24). The specific root exudates compounds (2-phenylglycine, L-homocysteine, D-asparagine, L-leucyl-L-alanine, β-1-Tomatine, and alpha-keto-gamma-(methylthio)butyric acid) on B24 chemotaxis and biofilm formation effects were further investigated (Figure 2G,H). The results indicated that with the exception of L-leucyl-L-alanine, the other five compounds significantly promoted both the chemotaxis and biofilm formation of B24. The effects of 2-phenylglycine, D-asparagine, and Beta1-tomatine on the chemotaxis and biofilm formation of B24 gradually increased with concentration, reaching significant levels at 0.5 mM. Furthermore, the chemotaxis and biofilm formation effects of L-homocysteine and alpha-keto-gamma-(methylthio)butyric acid on B24 exhibited concentration dependency, reaching significant levels at a concentration of 0.5 mM and then decreasing gradually.

### 3.5. Effects of B24 Inoculation on Plant Growth

Inoculation with B24 significantly enhanced various growth parameters of *L. chinensis*, including plant height, leaf length, leaf width, stem diameter, chlorophyll content, shoot fresh weight, root fresh weight, shoot dry weight, and root dry weight, which increased by 16.38%, 21.50%, 19.48%, 27.95%, 14.62%, 51.25%, 41.73%, 44.38%, and 22.38%, respectively (Figure 3). Mowing significantly reduced leaf width and stem diameter while significantly increasing the shoot fresh and dry weights. Under mowing treatments, inoculation with B24 significantly improved plant height, leaf length, chlorophyll content, shoot fresh weight, root fresh weight, shoot dry weight, and root dry weight, which increased by 38.02%, 19.02%, 20.61%, 42.73%, 74.75%, 55.28%, and 42.55%, respectively.

### 3.6. Effects of B24 Inoculation on the Physiological Indices of L. chinensis

Inoculation with B24 significantly increased the content of IAA, tZ, and SA in the *L. chinensis* roots by 26.33%, 162.94%, and 25.41%, respectively, while the GA3 content was significantly decreased by 12.00% (Figure 4A–C,F). Under mowing treatments, inoculation with B24 significantly increased the content of IAA, ABA, JA, and SA in the roots by 99.39%, 12.50%, 43.12%, and 124.86%, respectively, but significantly decreased tZ content (Figure 4A,B,D–F). These results indicate that under non-mowed treatments, B24 inoculation primarily enhanced the levels of plant hormones associated with growth and resistance (IAA, tZ, and SA), whereas under mowing treatments, it mainly increased the levels of plant hormones related to defense and damage regeneration (ABA, JA, and SA), thereby promoting the regeneration of *L. chinensis* after mowing. Additionally, inoculation with B24 significantly increased the TC, TN, TP, glucose, fructose, and sucrose contents in the roots by 15.23%, 16.36%, 156.74%, 8.27%, 21.36%, and 51.45%, respectively (Figure 4G–L). Mowing significantly reduced the TC, TN, glucose, fructose, and sucrose contents in the roots. Under mowing treatments, inoculation with B24 significantly increased the TN, TP, glucose, fructose, and sucrose content in the roots by 25.25%, 42.19%, 44.32%, 79.01%, and 43.71%, respectively. In summary, both B24 inoculation and B24 inoculation under mowing treatments significantly enhanced the TN, TP, glucose, fructose, and sucrose contents (Figure 4).

### 3.7. Characterization of Plant Metabolome in Response to B24 Inoculation

The DEMs were screened with VIP ≥ 1 and *p* < 0.05 as criteria. There were 52 DEMs (all upregulated), 95 DEMs (69 upregulated and 26 downregulated), and 103 DEMs (99 up and 4 down) detected in the BCK vs. B24, MCK vs. M24, and BCK vs. MCK comparison group, respectively (Figure 5A). The Venn diagram analysis showed that 30 DEMs were co-regulated among the different comparison groups (Figure 5B). The DEMs were classified and annotated into 10 main categories: amino acids and derivatives (58), lipids (1), carbohydrates and their derivatives (12), organic acids and their derivatives (11), organoheterocyclic compounds (9), phenolic acids (5), nucleotides and their derivatives (5), flavonoids (5), phytohormones (2), and other metabolites (8). Amino acids were the most abundant metabolites, followed by lipids, carbohydrates and their derivatives, and organic acids and their derivatives (Figure 5C). Almost all DEMs were significantly enriched with B24 inoculation, B24 inoculation under mowing treatments, and mowing treatments, among which, DEMs related to amino acids and carbohydrates were significantly enriched in all treatments (Figure 5D).

The VIP scores were used to screen important DEMs in order to identify key DEMs that respond to PGPR in promoting growth and adapting to stress. After B24 inoculation, mainly amino acids and carbohydrate DEMs changed, and the 10 DEMs with the highest VIP scores were all amino acids (Figure 5E,F). Additionally, amino acid DEMs were also significantly enriched with B24 inoculation after mowing, suggesting that they may play an important regulatory role in the growth promotion and regeneration of *L. chinensis*. Notably, L-Glutamate (L-Glu) and inositol galactoside were significantly upregulated with B24 inoculation with or without mowing. By mapping the DEMs to the KEGG database, the DEMs after B24 inoculation, mowing, and B24 inoculation under mowing treatments were mainly enriched in the pathways related to amino acid and carbohydrate metabolism (Appendix A).

### 3.8. Transcriptomic Analysis

Based on the Illumina sequencing platform, transcriptome sequencing was performed on 12 samples of *L. chinensis* roots, including RCK (non-inoculated B24 and non-mowed treatment), R24 (inoculated B24 treatment), MRCK (non-mowed but inoculated B24 treatment), MR24 (mowed and inoculated B24 treatment). A total of 142.27 Gb of Clean Data was obtained, the average sample data volume was 5.93 Gb, the GC content ranged from 50.94% to 55.95%, and the proportion of Q30 bases exceeded 91.84% (Appendix A). These results indicate that the sequencing data were of high quality and reliability, and were therefore suitable for subsequent analyses. Inoculation with B24 significantly altered the expression of 9261 DEGs, of which 4430 were upregulated and 4831 were downregulated. Under mowing treatments, B24 inoculation significantly changed the expression of 18,930 DEGs, with 12,791 upregulated and 6139 downregulated (Appendix A). The Venn diagram analysis showed that there were 46,401 DEGs across the three comparison groups (RCK vs. R24, MRCK vs. MR24, and RCK vs. MRCK), among which, 5482, 12,072, and 20,207 DEGs were unique to each comparison group, respectively, while 583 DEGs were common to all three groups, and 1997 DEGs were enriched under both the mowing and non-mowing treatments after B24 inoculation (Appendix A).

To explore the biological functions regulated by the DEGs, Gene Ontology (GO) and KEGG enrichment analyses were conducted. After inoculation with B24, the significantly enriched GO terms included catalytic activity, response to stimuli, organic acid biosynthetic process, and ATP binding (Appendix A). Carbohydrate metabolic process, cell wall, structural constituent of cell wall, ion transmembrane transporter activity, and transporter activity were significantly enriched after B24 inoculation with mowing (Appendix A). Additionally, mowing resulted in the significant enrichment of GO terms including catalytic activity, cell wall, structural constituent of cell wall, and transporter activity (Appendix A). After inoculation with B24, the significantly enriched KEGG pathways primarily included phenylpropanoid biosynthesis, linoleic acid metabolism, carbon metabolism, and nitrogen metabolism (Appendix A). Under mowing treatments, inoculation with B24 resulted in the significant enrichment of KEGG pathways related to carbohydrate metabolism, such as starch and sucrose metabolism and glycolysis/gluconeogenesis, as well as amino acid metabolism and energy metabolism (Appendix A). Furthermore, mowing resulted in the significant enrichment of KEGG pathways mainly associated with energy metabolism and amino acid metabolism (Appendix A). These findings indicate that inoculation with B24 primarily regulates pathways related to amino acid metabolism, carbohydrate metabolism, and energy metabolism.

### 3.9. Regulation of DEGs Related to Phytohormone Signaling

To investigate the effects of B24 inoculation on phytohormone signaling in *L. chinensis* roots, the expression of DEGs related to IAA, CTK, brassinosteroid (BR), ABA, JA, and SA was analyzed (Figure 6). In the IAA pathway, DEGs encoding auxin influx carrier (AUX1), gretchen Hagen 3 (GH3), small auxin-up RNA (SAUR) auxin synthesis. and response-related genes were significantly upregulated by B24 inoculation under mowing treatments. In the CTK pathway, the expressions of some DEGs encoding cytokinin response 1 (CRE1), histidine-phosphate transporter (AHP) and two-component response regulators (A-ARR and B-ARR) were significantly up-regulated with B24 inoculation under mowing treatments. In the BR pathway, B24 inoculation significantly upregulated the expression of three xyloglucan: xyloglucosyl transferase (TCH4) and 2 BR-signaling kinase (BSK) genes. The expression of two TCH4 genes was significantly upregulated after B24 inoculation under mowing treatments. In the ABA pathway, B24 significantly upregulated the expression of two DEGs encoding ABA receptor (PYR/PYL) and serine/threonine-protein kinase SRK2 (SnRK2). The expression of genes encoding PYR/PYL, protein phosphatase 2C (PP2C), and ABA responsive element binding factor (ABF) were significantly upregulated with B24 inoculation under mowing treatments. In the JA signaling pathway, B24 inoculation significantly downregulated the expression of most DEGs. In the SA pathway, the expression of two pathogenesis-related genes 1 (NPR1), three TGACG-binding factors (TGA), and two pathogenesis-related protein 1 (PR-1) genes were significantly upregulated with B24 inoculation under mowing treatments. Thus, B24 inoculation after mowing may have had a greater effect on the DEGs related to the plant hormone pathway, and promoted the regeneration of *L. chinensis* mainly by regulating interhormonal homeostasis.

### 3.10. Regulation of DEGs Related to Plant Growth and Development

Some transporters associated with nitrate, phosphate, and sugar were differentially expressed (Appendix A). Two sugar transport-related genes (SWEET12 and SWEET14) were significantly upregulated following B24 inoculation under both the mowing and non-mowing treatments, suggesting that B24 may enhance sugar transport in *L. chinensis*. Additionally, five ABC transport proteins (two ABCG36, ABCB11, and two ABCF3) associated with substance transporter were significantly upregulated after B24 inoculation under mowing treatments, with three of these DEGs also showing significant upregulation with B24 inoculation. Additionally, three phosphate transport-related genes (PHT1-11, PTH1-2, and PTH1-2) were significantly upregulated after B24 inoculation under both the mowing and non-mowing treatments (Appendix A). Furthermore, most DEGs involved in cell wall expansion, modification, and metabolism were significantly upregulated after B24 inoculation under both the mowing and non-mowing treatments, including EXPA3, EXLA2, XTH32, XTH26, BGLU44, CTL1, and BGLU26 (Appendix A).

### 3.11. Integrated Metabolomic and Transcriptomic Analysis to Explore Important Biological Pathways

According to the results of a previous KEGG enrichment pathway analysis of transcriptome and metabolome, B24 inoculation was found to mainly regulate the changes in DEGs and DEMs related to the pathways in amino acid metabolism, carbohydrate metabolism, and energy metabolism. The tricarboxylic acid cycle (TCA cycle) plays a central regulatory role in these metabolic pathways. Therefore, in this study, the TCA cycle, amino acid metabolism, carbohydrate metabolism, and nitrogen metabolism were regarded as key metabolic pathways for an in-depth analysis to explore the key biological mechanism of B24 in response to mowing stress and its role in promoting the growth of *L. chinensis* (Figure 7).

In the carbohydrate metabolism pathway, three DEMs and 57 DEGs were involved in the synthesis and decomposition of carbohydrates. Among them, the relative abundance of galactinol, melibiose, and sucrose increased significantly in three comparison groups (BCK vs. B24, MCK vs. M24, and BCK vs. MCK), and it was the highest in inoculation with B24 inoculation under mowing treatments. Most of the DEGs were significantly upregulated only with B24 inoculation under mowing treatments, including one sucrose synthase (SUS), three glycogen synthases (GYSs), three 1,4-α-glucan branching enzymes (GBE1s), two glycogen phosphorylases (PYGs), two α-amylases (AMYs), three phosphoglucomutases (PGMs), and three hexokinases (HKs). Additionally, regardless of whether there was a mowing treatment, the B24 inoculation significantly upregulated some DEGs, including insoluble isoenzyme (INV), two UTP-glucose-1-phosphate uridylyltransferases (UGP2), six β-glucosidases (bglX), and two raffinose synthases (RafS). These results indicate that the inoculation with B24 significantly regulated the changes in related DEGs and DEMs in the carbohydrate metabolism pathway, and the inoculation with B24 after mowing had the greatest impact on this pathway.

In the TCA cycle, most of the DEGs were significantly upregulated after inoculation with B24 both under the mowing and non-mowing treatments, including aconitate hydratase (ACO), isocitrate dehydrogenase (IDH1), succinate dehydrogenase (succinate dehydrogenase), and aconitate hydratase (ACO). Additionally, two ACOs, IDH1, one fumarate hydratase (fumD), and malate dehydrogenase (MDH1) were significantly downregulated after mowing, but were significantly upregulated with B24 inoculation under mowing treatments.

In the amino acid metabolism pathway, most DEGs were significantly upregulated with B24 inoculation after mowing, which included glutamic-oxaloacetic transaminase 1 (Got1), argininosuccinate synthase (argG), 4-aminobutyrate aminotransferase (ABAT), two 1-pyrroline-5-carboxylate dehydrogenases (putA), and pyrroline-5-carboxylate reductase (proC). Additionally, four DEGs encoding GOT1 and two DEGs encoding argG were significantly upregulated after B24 inoculation under both the mowing and non-mowing treatments. Some DEGs were significantly downregulated with mowing but were significantly upregulated with B24 inoculation under mowing treatments, such as argininosuccinate lyase (argH) and glutamate decarboxylase (GAD). Similar to the expression patterns of DEGs in this pathway, the number of DEMs significantly increased after B24 inoculation and B24 inoculation after mowing treatments. Among them, the relative abundances of L-Alanine, L-Aspartic acid, and 4-Aminobutyric acid were significantly increased after B24 inoculation. The relative abundances of L-Ornithine, L-Arginine, L-Asparagine, and L-Proline were significantly increased with B24 inoculation under mowing treatments. In addition, the relative abundance of L-Glutamic acid was significantly increased after B24 inoculation both under mowing and non-mowing treatments.

In the nitrogen metabolic pathway, most DEGs were significantly downregulated after mowing or B24 inoculation, but significantly upregulated with B24 inoculation after mowing treatments, such as nitrate/nitrite transporter (NRT), glutamate dehydrogenase (gdhA), and carbonic anhydrase (CA). Additionally, three glutamine synthetase (glnA) and one CA were significantly upregulated after inoculation with B24 in both mowing and non-mowing conditions. However, nitrate reductase (NR) and ferredoxin–nitrite reductase (nirA) were significantly downregulated after B24 inoculation and mowing.

### 3.12. qRT-PCR Analysis

To verify the accuracy of the transcriptome sequencing data, qRT-PCR was conducted to verify the expression of nine DEGs related to nitrate transport, phosphate transport, sugar transport, cell wall expansion, carbohydrate metabolism, and amino acid metabolism in *L. chinensis* roots. The results showed that the expression trend of seven DEGs in the four treatments was basically consistent with the transcriptome sequencing results, indicating that the transcriptome data were accurate and reliable (Figure 8).

## 4. Discussion

### 4.1. L. chinensis Recruits Key PGPR by Regulating Specific Root Exudates Under Overgrazing Stress

Plant root exudates play a pivotal role in plant–soil–microbe interactions, serving as both energy sources and recruitment signals for microorganisms [45]. Numerous studies have demonstrated that plants can selectively recruit beneficial rhizosphere soil microbes by releasing specific root exudates, such as amino acids, organic acids, and secondary metabolites in response to biotic and abiotic stress [46,47,48,49,50]. Consequently, under overgrazing, plants may also release specific root exudates to recruit beneficial microbes to the rhizosphere, thereby improving their stress adaptation capabilities [17]. To further validate this hypothesis, we analyzed the correlations between significantly enriched rhizosphere bacteria and key root exudates under overgrazing. The results revealed that the same compounds may exert different regulatory effects (positive or negative) on different bacteria, and the same bacteria may exhibit distinct nutritional preferences (Figure 2E,F). Similar findings have been reported in previous studies on the correlations between root exudates and rhizosphere microbes in Arabidopsis [51] and wheat [52]. Our study identified that *Paraburkholderia graminis*, a PGPR exclusively present under overgrazing, showed significant positive correlations with Beta 1-Tomatine, N′-Nitrosoanabasine, Asp-Glu, 5,7-Dihydroxy-3′,4′,5′-trimethoxyflavone, L-Homocystine, Chinese bittersweet alkaloid II, Lappaconitine, and D-Asparagine (Figure 2F). Therefore, after overgrazing, *L. chinensis* may recruit PGPR such as *Paraburkholderia graminis* to the rhizosphere by secreting primary metabolites, such as amino acids and organic acids, as well as secondary metabolites alkaloids. These recruited key PGPR further regulate the growth, development, and stress adaptation of the host plant, thereby alleviating grazing stress.

PGPR utilize chemotaxis to sense plant root exudate signals, migrate toward plant roots via flagella, and ultimately colonize on the root surface and forming biofilms [53]. Numerous studies have demonstrated that amino acids and organic acids in root exudates, serving as carbon sources, nitrogen sources, and signaling molecules for microbes, can induce PGPR chemotaxis and enhance their rhizosphere colonization and biofilm formation capabilities. For instance, root exudate benzoic acid can promote the recruitment of *Paraburkholderia* in gramineous crops (e.g., sugarcane and maize), thereby enhancing nitrogen fixation [14,54]. Additionally, compounds such as flavonoids have been found to influence the chemotaxis and biofilm formation of *Paraburkholderia xenovorans* LB400, affecting its colonization on plant roots [55]. Soybean root exudates significantly promote biofilm formation and rhizosphere colonization by *B. amyloliquefaciens* BNM339 [56]. Therefore, we selected significantly enriched amino acids, alkaloids, and organic acids root exudates to investigate their effects on the chemotaxis and biofilm formation of *Paraburkholderia graminis*.

Alkaloids, as a component of root exudates, may play a role in signal transduction and microbial attraction, but their specific mechanisms of action are limited. It has been reported that alkaloids secreted by tea plants may influence rhizosphere microbial communities, but there is no evidence confirming whether these alkaloids can recruit PGPR [57]. Our findings align with previous studies, demonstrating that not only specific amino acids and organic acids but also alkaloids such as Beta 1-Tomatine significantly enhance the chemotaxis and biofilm formation of B24 in a concentration-dependent manner. These compounds act as rhizosphere signaling molecules or carbon sources, attracting potentially beneficial rhizosphere bacteria and improving the rhizosphere environment to enhance stress adaptation. Additionally, although 2-Phenylglycine showed no significant correlation with *Paraburkholderia graminis*, it still significantly promoted the chemotaxis and biofilm formation of B24. These results indicate that different root exudates have distinct chemotactic effects on bacterial strains, and different strains exhibit distinct nutritional preferences [30]. These specific root exudates can serve as both nutrient sources and chemical attractants for rhizosphere microorganisms assembling and shaping the rhizosphere microbial community. Through chemotaxis and biofilm formation, they initiate communication between plants and bacteria, mediating bacterial colonization and symbiotic interactions with plant roots [45,58]. In summary, under overgrazing, *L. chinensis* may employ a “cry for help” strategy by releasing specific root exudates (2-phenylglycine, L-homocysteine, D-asparagine, β-1-Tomatine, and Alpha-keto-gamma-(methylthio)butyric acid) to actively seek assistance from beneficial microbes such as *Paraburkholderia graminis*, thereby enhancing its adaptability to overgrazing stress.

### 4.2. PGPR Inoculation Promoted the Growth and Regeneration of L. chinensis After Mowing by Promoting Nutrient Absorption, Transport, and Cell Wall Expansion

PGPR promote plant growth through phosphorus solubilization, nitrogen fixation, phytohormone production, and systemic resistance induction [59]. They can also increase photosynthetic efficiency, and chlorophyll content [60]. Although PGPR inoculation promotes growth in Poaceae family plants, such as maize [61], wheat [62], rice, and sugarcane [63], there are few studies on the growth-promoting effects and regulatory mechanisms of PGPR on *L. chinensis*. Our study showed that PGPR B24 inoculation significantly promotes the growth and regeneration of *L. chinensis* after mowing (Figure 4), which was similar to the results in previous studies [63]. Previous studies have shown that PGPR-induced root development, increased aboveground biomass, and plant growth promotion are associated with IAA synthesis [64]. Furthermore, nitrogen-fixing bacteria and phosphate-solubilizing bacteria could improve seed germination rates, root and shoot biomass, and overall plant growth [65]. Alam et al. (2001) found that inoculation with *Azotobacter* sp. significantly increased nitrogen accumulation, rice dry matter, and yield [66]. Therefore, the enhanced regeneration capacity of *L. chinensis* observed after PGPR inoculation may be attributed to its beneficial growth-promoting traits, such as IAA secretion, nitrogen fixation, and phosphate solubilization.

Grassland plants adapt to grazing or mowing through shoot regeneration, with the reallocation of nitrogen and phosphorus from roots to shoots serving as a critical nutrient utilization strategy to support regeneration [12,67]. The TC, TN, glucose, fructose, and sucrose contents in *L. chinensis* roots decreased significantly after mowing, while they increased with B24 inoculation under the mowing and non-mowing treatments (Figure 4). Furthermore, under both the mowing and non-mowing treatments, B24 inoculation significant upregulated several ABC transporters, nitrate transporters (NRTs), phosphate transporter genes, and sugar transporter genes. Previous studies have demonstrated that the application of PGPR promotes nitrogen acquisition and transport in roots by significantly upregulating the NRT gene family. Our results indicate that inoculation with B24 enhances nutrient uptake and transport in the roots, providing sufficient nutrient support for shoot regeneration under mowing treatments, which aligns with previous research [32,68]. There are extensive studies showing that inoculation with PGPR significantly regulates plant growth, enhances plant nutrient use efficiency, and improves metabolite regulation [13,69]. Additionally, B24 enhances sugar accumulation by promoting sugar transport, thereby increasing the chlorophyll content and photosynthetic efficiency. This further promotes the growth and post-mowing regeneration of *L. chinensis* and helps the plant adapt to environmental stress.

PGPR can modulate the expression of functional genes associated with plant growth and development through the production of phytohormones and the mobilization of nutrients [14]. In this study, we found that inoculation with B24 significantly promoted the growth of *L. chinensis* and systematically modulated the transcription of key genes involved in plant growth and development. In particular, most of the DEGs involved in cell wall expansion, modification and metabolism were significantly upregulated after B24 inoculation regardless of mowing or non-mowing treatments (Appendix A), thus promoting cell wall relaxation, expansion, and extension, promoting cell division and enhancing cell plasticity, and ultimately promoting plant growth and regeneration after mowing.

### 4.3. PGPR Inoculation Responds to Mowing of L. chinensis by Regulating Plant Hormone Signaling Pathways

Plant hormones are important signaling molecules in plants that regulate plant growth, development, and physiological processes. PGPR also actively regulate various physiological processes by controlling endogenous plant hormones, nutrient balance, and plant signaling molecules, thereby participating in plant growth [70]. Studies have shown that PGPR promote plant growth or enhance plant defenses by regulating the homeostasis of plant hormones [71,72]. Our results indicated that under the mowing treatments, inoculation with B24 regulated pathways related to growth, such as IAA, CTK, and BR, as well as stress resistance and defense-related pathways such as ABA and SA in *L. chinensis* roots (Figure 6). This helped to maintain hormonal homeostasis and enabled the plant to alleviate mowing stress. Similarly, under mowing treatments, B24 inoculation significantly increased the contents of IAA, ABA, and SA in *L. chinensis* roots (Figure 4).

In the IAA and CTK signaling pathways, most genes encoding AUX1, GH3, SAUR, CRE1, AHP, A-ARR, and B-ARR were significantly upregulated after B24 inoculation under the mowing treatments, thereby regulating the growth and development processes of *L. chinensis*. It has been shown that PGPR inoculation can regulate gene expression related to growth and development, nutrient uptake and transport, and plant hormone signaling, particularly the IAA response pathway [14]. PGPR inoculation affects root development by synthesizing IAA or regulating the distribution and transport of IAA in plant tissues, thereby increasing root surface area, aboveground biomass, and nutrient uptake capacity [73]. Previous studies have demonstrated that IAA and CTKs play a key role in promoting the regeneration of winter wheat and barley after mowing [74]. Our findings revealed that under the mowing treatments, the majority of genes associated with PYR/PYL, PP2C, and ABF were upregulated after B24 inoculation, suggesting the activation of the ABA signaling pathway, which was consistent with previous studies [75]. Furthermore, most genes encoding NPR1, TGA, and PR-1 were significantly upregulated after B24 inoculation under the mowing treatments, suggesting that SA signaling may also be involved in the interaction between *L. chinensis* and PGPR B24 after mowing. Our results confirmed that B24 helps plants achieve a balance between defense and growth signaling pathways, enhances nutrient uptake and transport, and significantly promotes the regeneration and overall recovery of *L. chinensis* after mowing. In summary, B24 regulates the homeostasis of endogenous plant hormones and plays a key role in the growth and post-mowing regeneration of *L. chinensis* through the synergistic effects of hormones and the trade-off between growth and defense.

### 4.4. PGPR Inoculation Responds to Mowing Stress by Regulating Amino Acid Metabolism, Energy Metabolism, and Carbohydrate Metabolism of L. chinensis

During the post-mowing recovery phase, plants typically require rapid carbohydrate synthesis to facilitate regeneration. PGPR can expedite this recovery process by enhancing photosynthetic efficiency, optimizing carbohydrate metabolism, and promoting sugar biosynthesis [76]. Starch and sucrose metabolism is the main carbohydrate metabolic pathway in *L. chinensis* roots. During starch synthesis, most genes encoding glycogen synthase (GYS), 1,4-α-glucan branching enzyme (GBE1), two glycogen phosphorylases (PYG), α-amylase (AMY), phosphoglucomutase (PGM), and hexokinase (HK) were significantly upregulated after B24 inoculation under mowing treatments (Figure 7A). These findings demonstrate that B24 enhances the conversion of ADP-glucose into starch by modulating the expression of these DEGs, thereby accelerating the accumulation of starch in *L. chinensis* roots after mowing. Sucrose is the primary form of photosynthetic product that is stored and transported in plants. Sucrose synthase (SUS) as a key enzyme in sucrose metabolism, plays a central role in the reversible degradation and synthesis of sucrose, thereby regulating energy storage and maintaining metabolic homeostasis [77]. Additionally, sucrose is converted into fructose and glucose through the catalysis of insoluble isozymes (INV), SUS, UTP-glucose-1-phosphate uridylyltransferase (UGP2), and PGM, thus regulating plant sugar metabolism. Most of these genes exhibited significant upregulation following B24 inoculation under both mowing and non-mowing treatments, indicating an enhanced capacity for sucrose conversion into glucose and fructose. This process facilitates the accumulation of respiratory substrates, thereby promoting root growth, above-ground biomass production, and post-mowing regeneration [78]. Consequently, the B24 strain plays a key regulatory role in sucrose metabolism in *L. chinensis*. These results are consistent with the significant accumulation of sucrose content after B24 inoculation (Figure 4L).

The TCA cycle is a crucial component of plant respiration, providing energy and carbon skeletons for amino acid synthesis [79]. Mowing caused a significant downregulation of key TCA cycle enzymes (e.g., ACO, IDH1, fumD, and MDH1), while their expression was significantly upregulated after B24 inoculation under mowing treatments (Figure 7B). This may be due to the suppression of energy metabolism after the plant was subjected to mowing stress, which subsequently led to plants preferentially allocating resources to other stress response pathways (e.g., antioxidant, wound healing, or signaling). However, inoculation with B24 significantly promoted the upregulation of these genes and restored the activity of TCA cycle, promoted the acceleration of energy metabolism and the accumulation of intermediate metabolites, thus providing material and energy support for cell repair and anti-stress response. This ultimately enabled plants to better cope with mowing stress.

L-Aspartate (L-asp), an important metabolic hub associated with multiple pathways, plays a pivotal role in plant energy supply, growth regulation, and stress responses. After B24 inoculation, the relative abundance of L-asp significantly increased, which was consistent with previous studies [80]. Similarly, the relative abundance of L-asparagine (L-Asn) significantly increased after B24 inoculation under mowing treatments. The AsnB gene encodes asparagine synthetase, an enzyme that catalyzes the synthesis of L-Asn from L-Asp and glutamine (Gln). Our results indicated that several asnB genes were significantly downregulated after mowing or B24 inoculation. Consequently, the degradation of L-asp was inhibited, indicating significant accumulation of L-asp after mowing or B24 inoculation. This finding was aligned with the significant increase in the relative abundance of L-Asp observed in the metabolomics data after B24 inoculation (Figure 7C). In amino acid synthesis, glutamate acts as a nitrogen donor and can be converted into γ-aminobutyric acid (GABA) via glutamate decarboxylase (GAD), playing a crucial role in balancing carbon and nitrogen metabolism, as well as regulating plant growth, development, and defense responses [81]. In this study, most genes encoding GAD (the primary enzyme for GABA synthesis) were significantly downregulated after mowing but were significantly upregulated after B24 inoculation under mowing treatments (Figure 7C). This suggests that *L. chinensis* may prioritize other stress response pathways after mowing, temporarily reducing the demand for GABA synthesis. Thus, B24 inoculation may induce stress resistance in plants by promoting GABA synthesis to help the plant better cope with mowing stress. Additionally, B24 may enhance GAD expression by influencing plant signaling pathways, such as the GABA pathway or stress response pathways. This is similar to other studies that have shown GABA synthesis in PGPR-inoculated tomatoes in response to abiotic stress and promote plant growth [75,80]. Moreover, GABA alleviates stress and enhances growth in tomatoes by altering GAD gene expression and regulating amino acid synthesis and TCA cycle [82]. Additionally, our results also show that the gene encoding glutamate oxaloacetate transaminase (GOT1) was significantly upregulated after B24 inoculation under both the mowing and non-mowing treatments. This gene encodes aspartate aminotransferase (AAT), a bridge between carbon and nitrogen metabolism, catalyzing the conversion of oxaloacetate and glutamate into α-ketoglutarate and aspartate, linking amino acid metabolism with the TCA cycle [83]. After inoculation with B24, nitrogen assimilation pathways may be activated, improving the nitrogen utilization efficiency. This enhances GOT1 expression, further optimizing amino acid metabolism and allowing α-ketoglutarate to enter the TCA cycle, providing more energy for the plant [84]. After mowing, plants need to rapidly mobilize energy and amino acid metabolism to repair damage and restore growth. The significant upregulation of GOT1 after B24 inoculation suggests it may alleviate mowing stress by promoting amino acid metabolism and energy supply.

In the nitrogen metabolic pathway, glutamine synthetase (glnA) and carbonic anhydrase (CA) were significantly upregulated after B24 inoculation under both mowing and non-mowing treatments (Figure 7D). glnA is a key gene responsible for nitrogen assimilation and redistribution, converting ammonia (NH3) absorbed by plant roots and ammonia produced by amino acid decomposition into Gln and glutamate [81]. This indicates that nitrogen metabolism is regulated in plants after B24 inoculation under both mowing and non-mowing treatments, which aligned with previous studies [85]. Additionally, L-Glu as a hub of nitrogen metabolism, plays an indispensable role in regulating plant growth, stress resistance, and rhizosphere ecosystems. Its relative abundance significantly increased after B24 inoculation under both the mowing and non-mowing treatments (Figure 7). This suggests that the B24 strain plays a key role in improving plant nitrogen metabolism efficiency, environmental adaptability, stress resistance, and amino acid metabolism regulation. Therefore, B24 inoculation promotes ammonia assimilation, reduces the need for nitrate reduction, optimizes nitrogen metabolism in *L. chinensis*, and enhances plant growth, recovery, and stress resilience under mowing stress. In summary, PGPR B24 coordinates the metabolic cycles of amino acids, enhancing the synergy between nitrogen and carbon metabolism, and improving plant metabolic activity and stress resistance.

## 5. Conclusions

This study illustrates how *L. chinensis* adapts to overgrazing stress by recruiting specific PGPR. *L. chinensis* with overgrazing history to alleviate grazing stress by regulating specific root exudates (e.g., amino acids, alkaloids, and organic acids) to recruit the PGPR *Paraburkholderia graminis* (B24). Additionally, B24 inoculation promotes physiological and phenotypic changes in the plant and systematically regulates the transcription of the key genes involved in plant growth and development, such as hormone signaling, nutrient transport, and cell wall modification and expansion. This enhances carbon and nitrogen metabolism, amino acid metabolism, and energy supply in *L. chinensis* after mowing, synergistically improving its growth, development, and post-mowing regeneration capacity. These findings showed that plants under overgrazing stress increase the recruitment of beneficial microbes to the rhizosphere and deepen our understanding of plant–microbe synergies in grazing grassland ecosystems. Collectively, these findings provide novel insights that could help to develop new fertilizers and better management approaches for degraded grassland restoration and sustainable grassland development.

## Figures and Tables

**Figure 1 microorganisms-13-01225-f001:**
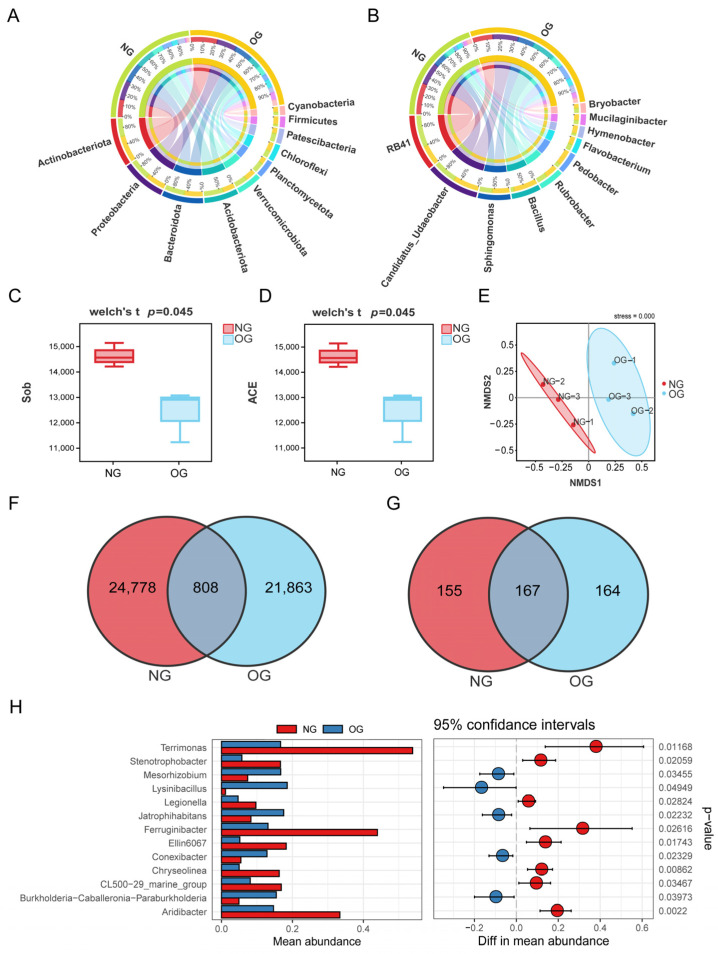
Effects of overgrazing on the rhizosphere bacterial community of *L. chinensis*. Circos maps of species distribution at the phylum level (**A**) and genus level (**B**). α-diversity analysis based on Sobs (**C**) and ACE (**D**) index. (**E**) Non-metric multidimensional scaling (NMDS) analysis based on Bray–Curtis distance. Each treatment contains three repetitions. (**F**) Venn diagram analysis of common and unique bacterial amplicon sequence variants (ASVs) under the overgrazed (OG) treatments. (**G**) Venn diagram analysis of common and unique bacterial species under overgrazing. (**H**) Differentially enriched bacterial genera under overgrazing were identified based on a Welch’s *t*-test.

**Figure 2 microorganisms-13-01225-f002:**
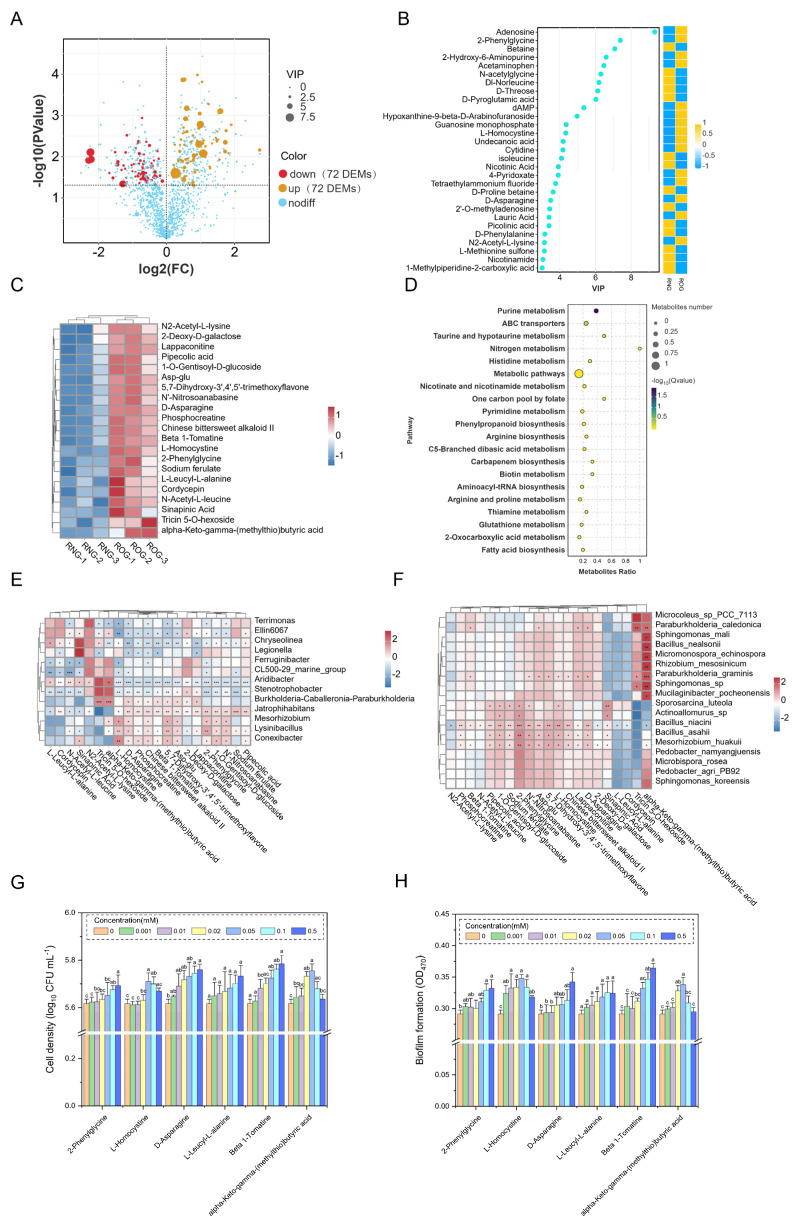
Interaction between specific root exudates and key PGPR after overgrazing. (**A**) Volcano plot of DEMs. Dots represent individual DEMs with fold change on the x-axis and statistical significance on the y-axis. Horizontal dashed line indicates the *p*-value threshold. (**B**) VIP bar plot of DEMs (VIP ≥ 3, *p* < 0.05). Each dot represents a metabolite. The dot’s position on the axis (usually x-axis) shows the VIP score of that metabolite, calculated from a PLS-DA model. (**C**) Cluster heatmap of significantly enriched root exudate under overgrazing. (**D**) KEGG enrichment bubble plot of the DEMs (Top 20). Correlation analysis between key root exudates and rhizosphere bacteria genus (**E**) and species (**F**). The symbols *, ** and *** indicates *p* < 0.05, *p* < 0.01 and *p* < 0.001, respectively. Effects of different concentrations of root exudate compounds on the chemotaxis (**G**) and biofilm formation (**H**) of strains B24. Different lowercase letters indicate significant differences between treatments (*p* < 0.05).

**Figure 3 microorganisms-13-01225-f003:**
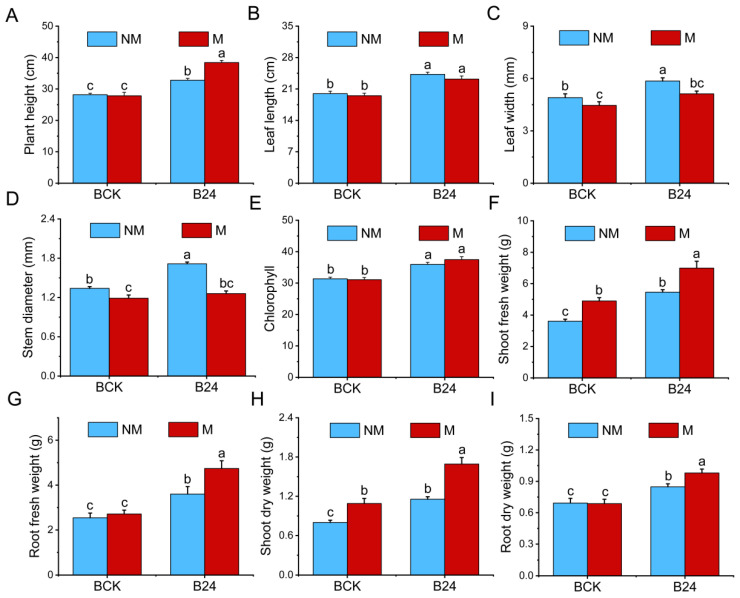
Effects of B24 inoculation on the growth indexes of *L. chinensis*. (**A**–**I**) Represent plant height, leaf length, leaf width, stem diameter, chlorophyll content, shoot fresh weight, root fresh weight, shoot dry weight, and root dry weight, respectively. The data are shown as mean ± standard error. Different lowercase letters indicate significant differences between different treatments (*p* < 0.05).

**Figure 4 microorganisms-13-01225-f004:**
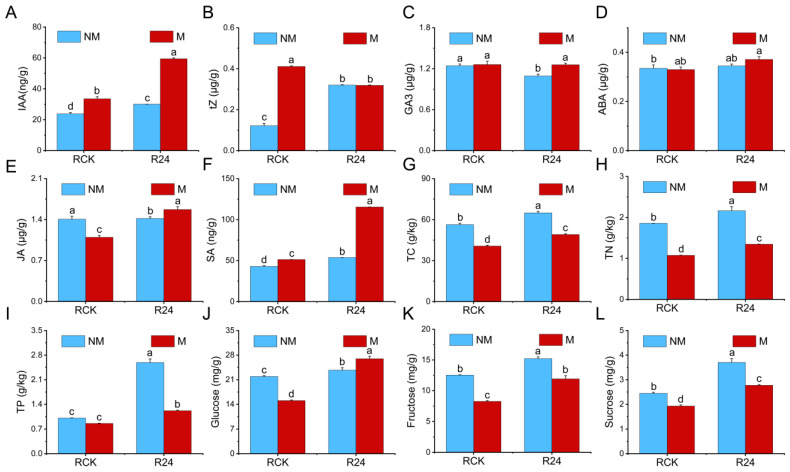
Effects of B24 inoculation on the physiological indices of *L. chinensis*. (**A**–**L**) Represent acetic acid (IAA), zeatin (tZ), gibberellin (GA3), abscisic acid (ABA), jasmonic acid (JA) and salicylic acid (SA), total carbon (TC), total nitrogen (TN), total phosphorus (TP), glucose, fructose, and sucrose, respectively. Different lowercase letters indicate significant differences between different treatments (*p* < 0.05).

**Figure 5 microorganisms-13-01225-f005:**
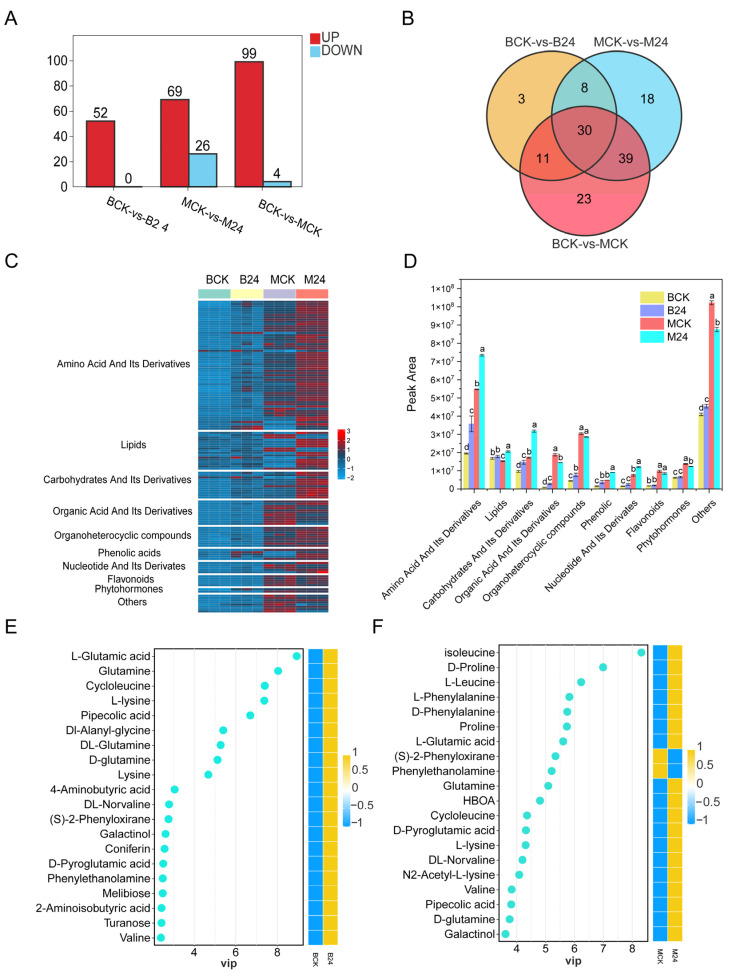
Changes in root DEMs under different treatments. (**A**) Numbers of DEMs under different treatments. (**B**) Venn diagram of DEMs under different treatments. (**C**) Heatmap of root DEMs abundances in different treatments groups. (**D**) Peak area (abundance) of different DEMs categories. A variable importance in the projection (VIP) bar plot of key DEMs (VIP top20) among BCK vs. B24 (**E**) and MCK vs. M24 (**F**). Each dot represents a metabolite. The dot’s position on the axis (usually x-axis) shows the VIP score of that metabolite, calculated from a PLS-DA model. Different lowercase letters indicate significant differences between different treatments (*p* < 0.05).

**Figure 6 microorganisms-13-01225-f006:**
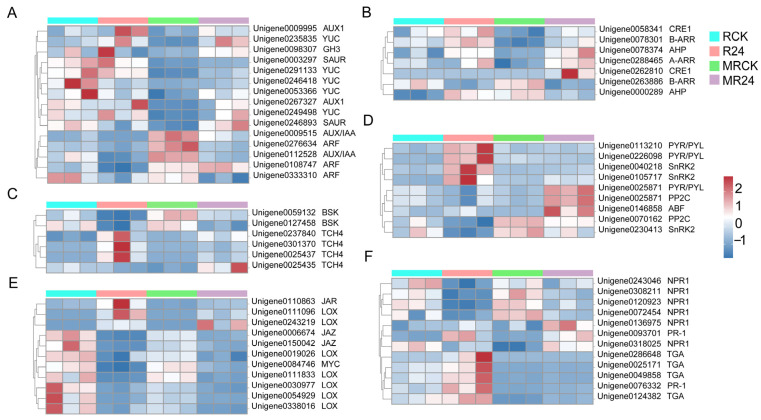
Expression of DEGs involved in phytohormone signaling. (**A**–**F**) Expression of DEGs related to auxin (IAA), cytokinin (CTK), brassinolide (BR), abscisic acid (ABA), jasmonic acid (JA), and salicylic acid (SA) pathways.

**Figure 7 microorganisms-13-01225-f007:**
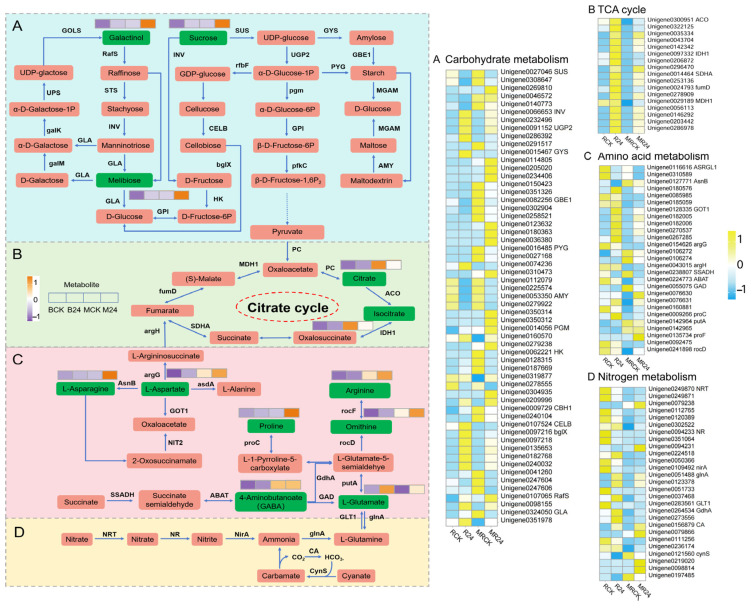
Diagram of important metabolic pathways combined with transcriptome and metabolome association analysis. (**A**) carbohydrate metabolism. (**B**) The citric acid cycle. (**C**) Amino acid metabolism. (**D**) Nitrogen metabolic.

**Figure 8 microorganisms-13-01225-f008:**
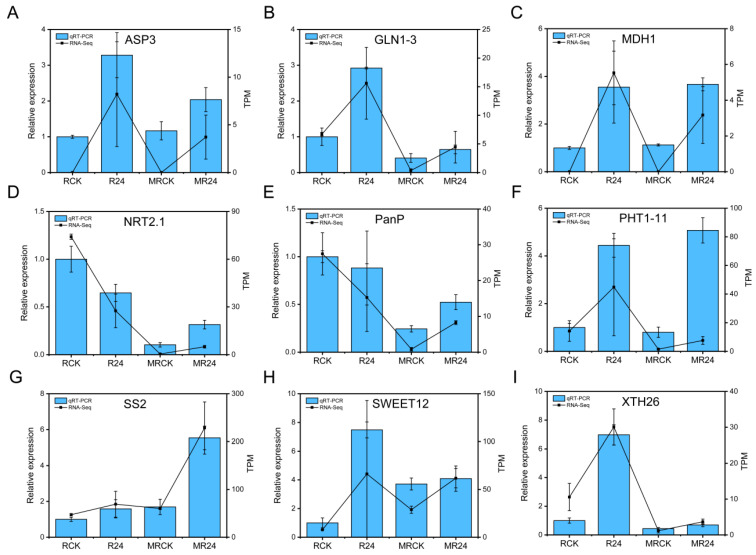
Real-time quantitative PCR analysis of related DEGs in *L. chinensis* roots under different treatments. (**A**–**I**) Represent the relative expression of ASP3, GLN1-3, MDH1, NRT2.1, PanP, PHT1-11, SS2, SWEET12, XTH26.

## Data Availability

All raw sequencing data have been submitted to the National Center for Biotechnology Information (NCBI) Sequence Read Archive (SRA) database under the accession number PRJNA1114925 and PRJNA1116349.

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
