# Peer review of "Alleviating Overgrazing Stress and Promoting Grassland Plant Regeneration via Root Exudate-Mediated Recruitment of Beneficial Bacteria"

_microorganisms, 2025, doi:10.3390/microorganisms13061225_

Round 1
Reviewer 1 Report
Comments and Suggestions for Authors
Dear authors,
The manuscript is interesting in terms of subject matter and of interest to those involved in the grassland field. However, some clarifications and corrections are necessary to increase the value of your work. These can be found below and also directly in the manuscript.
Line 15: Related to: <... the key PGPR Paraburkholderia graminis (B24) by regulat... >
Distinguish between PGPR and the species in question. From what you have said here, it is not clear whether you are referring to a group of bacteria that promote growth or to Paraburkholderia graminis?
Line 81: Please write Fusarium oxysporum in italics
Line 82: Please write Bacillus amyloliquefaciens in italics
Line 68: In the abstract you talk about the bacterium Paraburkholderia graminis. However, nowhere in the Introduction do you refer to this key element. That is, you should describe it and its specific implications for plant growth.
Line 111: Related to: <... to identify core rhizosphere PGPR...)
Please mention which PGPR species you had in mind when starting your research because it is too general. Emphasize the specificity more and avoid using the group almost throughout the paper (as I have observed).
Line 129: Please write Leymus chinensis in italics
Line 134: Please write Leymus chinensis in italics
Line 120-122: At 2.1 Study site and sampling: Place the study in time. In what years were the observations made? It is not clear. You only refer to the annual average precipitation from 2012 to 2018 but do not mention when you made the observations related to the study. In my opinion, this phrase about precipitation is not relevant; it seems that you do not associate it with any real activity. Or explain what the level of precipitation is for in your study and why you chose the period 2012-2018?
Line 253: Related to: <... Candidatus_ Udaeobacter ...Please write Candidatus udaeobacter and also in italics
Line 833: The name of the journal is: Agricultural and Forest Meteorology
Line 858: The name of the journal is: Trend in Plant Science.
Line 866: The name of the journal is: Nature Communication
Line 864: The name of the journal is: Journal of Experimental Botany
Line 883: The name of the journal is: Soil Biology and Biochemistry
Line 896: The name of the journal is: Plant Physiology
Line 899: The name of the journal is: FEMS Microbiology Ecology
Line 901: The name of the journal is: Molecular Microbiology
Line 916: The name of the journal is: Nucleic Acids Research
Line 917: The name of the journal is: BMC Bioinformatics
Line 919: The name of the journal is: Genome Biology
Line 924: The name of the journal is: The ISME Journal
Line 943: The name of the journal is: Nature Reviews Microbiology
Line 949: The name of the journal is: Current Microbiology
Line 955: The name of the journal is: Molecular Plant
... And so on...please check all references regarding the correct name of the journal.
Kind regards,
R
Author Response
Dear Reviewer,
On behalf of my co-authors, we thank you very much for giving us an opportunity to revise our manuscript. We thank the reviewers for the thoughtful and valuable comments and suggestions on the manuscript. We have studied the comments carefully and have made revision accordingly (The manuscript is marked in red font). In addition, we discovered some other problems in the manuscript, which we have also revised. The major changes are described in detail in the attached point-by-point answers for your further evaluation. The comments are listed below in black font, and our responses are provided in blue font.
We would like to express our great appreciation to you and reviewers for comments on our paper. Looking forward to hearing from you. Thank you and best regards.
Sincerely,
Weibo Ren
School of Ecology and Environment
Inner Mongolia University
On behalf of all authors.
Reviewer 1
The manuscript is interesting in terms of subject matter and of interest to those involved in the grassland field. However, some clarifications and corrections are necessary to increase the value of your work. These can be found below and also directly in the manuscript.
Comments 1: Line 15: Related to: <... the key PGPR Paraburkholderia graminis (B24) by regulat... >
Response 1: Thank you for your comments. We have now revised the sentence (Page 1, Line 14) to clearly reveal the specific bacterial strain described in our study.
Comments 2: Distinguish between PGPR and the species in question. From what you have said here, it is not clear whether you are referring to a group of bacteria that promote growth or to Paraburkholderia graminis?
Response 2:Thank you for pointing this out. We agree that the original sentence was ambiguous. We have now revised the sentence (Page 1, Line 14) to clearly reveal the specific bacterial strain described in our study.
Comments 3: Line 81: Please write Fusarium oxysporum in italics
Response 3: Thank you for pointing this out. We have revised the Fusarium oxysporum in italics on Page 2, Lines 83-84.
Comments 4: Line 82: Please write Bacillus amyloliquefaciens in italics
Response 4: Thank you for pointing this out. We have revised the Bacillus amyloliquefaciens in italics on Page 2, Line 85.
Comments 5: Line 68: In the abstract you talk about the bacterium Paraburkholderia graminis. However, nowhere in the Introduction do you refer to this key element. That is, you should describe it and its specific implications for plant growth.
Response 5: Thank you for your helpful suggestion. We have added the relevant descriptions about the Paraburkholderia to promote plant growth. Meanwhile, we have also deleted and revised the relevant content about the impact of PGPR on plant growth. Detailed modification details can be found in Page 2, Lines 58-66.
Comments 6: Line 111: Related to: <... to identify core rhizosphere PGPR...)
Response 6: Thank you for your comments. We have revised the relevant descriptions on Page 3, Line 115 and 117.
Comments 7: Please mention which PGPR species you had in mind when starting your research because it is too general. Emphasize the specificity more and avoid using the group almost throughout the paper (as I have observed).
Response 7: Thank you for your valuable suggestion. We have added the relevant descriptions on Page 2, Lines 60-65 and Page 3, Line 115 and 117.
Comments 8: Line 129: Please write Leymus chinensis in italics
Response 8: Thank you for pointing this out. We have revised the Leymus chinensis in abbreviation and italics on Page 3, Line 133.
Comments 9: Line 134: Please write Leymus chinensis in italics
Response 9: Thank you for pointing this out. We have revised the Leymus chinensis in italics on Page 3, Line 138.
Comments 10: Line 120-122: At 2.1 Study site and sampling: Place the study in time. In what years were the observations made? It is not clear. You only refer to the annual average precipitation from 2012 to 2018 but do not mention when you made the observations related to the study. In my opinion, this phrase about precipitation is not relevant; it seems that you do not associate it with any real activity. Or explain what the level of precipitation is for in your study and why you chose the period 2012-2018?
Response 10: Thank you for your valuable comment. We have revised the text to clearly indicate that the rhizosphere soil sampling was conducted during the peak growing season in July 2022 (Page 3, Lines 124-126). We analyzed the precipitation data from 2012 to 2022 in the study area and found that rainfall from May to September accounted for 83.2% of the annual total (Page 3, Line126). Referring to the average annual precipitation during this period aims to characterize the long-term climatic background of the study site. This time frame was chosen because it reflects a relatively stable period in local precipitation trends based on meteorological records.
Comments 11: Line 253: Related to: <... Candidatus_ Udaeobacter ...Please write Candidatus udaeobacter and also in italics
Response 11: Thank you for pointing this out. We have revised the Candidatus_ Udaeobacter in italics on Page 6, Line 272.
Comments 12:
Line 833: The name of the journal is: Agricultural and Forest Meteorology
Line 858: The name of the journal is: Trend in Plant Science.
Line 866: The name of the journal is: Nature Communication
Line 864: The name of the journal is: Journal of Experimental Botany
Line 883: The name of the journal is: Soil Biology and Biochemistry
Line 896: The name of the journal is: Plant Physiology
Line 899: The name of the journal is: FEMS Microbiology Ecology
Line 901: The name of the journal is: Molecular Microbiology
Line 916: The name of the journal is: Nucleic Acids Research
Line 917: The name of the journal is: BMC Bioinformatics
Line 919: The name of the journal is: Genome Biology
Line 924: The name of the journal is: The ISME Journal
Line 943: The name of the journal is: Nature Reviews Microbiology
Line 949: The name of the journal is: Current Microbiology
Line 955: The name of the journal is: Molecular Plant
... And so on...please check all references regarding the correct name of the journal.
Response 12: Thank you for your valuable suggestion. We have modified the name of the journal mentioned above and thoroughly check all references regarding the correct name of the journal.

Reviewer 2 Report
Comments and Suggestions for Authors
The manuscript presents a study focused on two main aspects: a) the bacterial diversity associated with Leymus chinensis under no grazing and overgrazing conditions, and b) the effects of Paraburkholderia graminis B24 on the transcriptomic and metabolomic profiles of Leymus chinensis under mown and unmown conditions.
However, the overall storyline is unclear and insufficiently developed. It appears that the authors are attempting to combine two distinct studies within a single manuscript, but there is a lack of coherence and integration between them. This disconnection—or possibly a lack of background information—hinders the reader's ability to grasp the aims and implications of the study entirely.
Although the manuscript provides potentially valuable descriptive insights at the transcriptomic and metabolomic levels, the methodology section lacks organization and appears incomplete, which makes the experimental workflow difficult to follow.
The topic is interesting, and the manuscript has potential for publication, but major revisions are required to improve the study's clarity, coherence, and completeness.
General comments
Please adjust the manuscript to the requirements of the Microorganism MDPI journal. According to the journal's requirements, it lacks information on the corresponding author and email address.
Please write in italics the scientific names of microorganisms, plants, and genes.
Throughout the article, separate the text from the square brackets in the references. Although only a few examples are mentioned, this should be corrected throughout the article.
Changes
In Line 71. Please change "lay" for "play"
Lines 81 and 82. Scientific names must be written in italics; please review them in all text of the manuscript.
Line 93. Azotobacter genus in italics and capital letters. Please review scientific names in the manuscript.
Line 96. Please separate "stress" from the square bracket
Line 106: Please delete "remains"
Lines 145-153. This section lacks coherence with the information presented in Section 2.1. In section 2.2, the isolation and characterization of Paraburkholderia graminis B24 is introduced directly, without proper context. Please provide additional information to clarify why isolate B24 was selected and why the focus was explicitly placed on Paraburkholderia.
Line 158-160: For reagents, please include CAS number, company, or reagent brand.
Line 185. Please separate "stress" from the square bracket
Line 191. Please express the colony-forming units per milliliter (CFU/mL) in exponential notation.
Line 205-214. Where are the results that describe the influence of Paraburkholderia graminis B24 on the community structure of Leymus chinensis?
Line 231-240. It should be specified that the exudates analyzed were those released by the plants collected in section 2.1. Additionally, the distinction should be made between these exudates and the metabolites produced in the Leymus chinensis roots due to the effect of Paraburkholderia graminis B24.
Line 249-266. This lacks consistency or coherence with the culturable bacteria described in lines 267–272.
Line 272-280. Please specify whether the information shown in Figure 1 corresponds to 16S rRNA sequencing culturable rhizosphere bacteria from Leymus chinensis or is based on metagenomic analysis. Alternatively, clarify how this data was obtained.
Line 281-289. In the same context, is section 3.2 connected to section 3.1? or are they addressing different topics? Or does it refer to culturable microorganisms?
Line 269 and 296. Some parts of the text are redundant.
Line 290-302. Are these results related to the data presented in section 2.1? (line 134-144)
Line 293, 294, and 297. Is referring to Figures 1A, 1B, 1C, and 1D correct, or did you mean Figures 2A, 2B, 2C, and 2D?
Line 303-318. How did they obtain the correlation? What are they analyzing? Are they culturable or non-culturable bacteria?
Figure 2A. Is it correct to say Down (72 genes) and Up (72 genes), or perhaps does it refer to differentially synthesized metabolites?
Line 326-329. Are these compounds the most commonly synthesized in overgrazing? Or why were they chosen?
Line 389-390. What is the meaning of (RCK vs R24, MRCK vs MR24, and RCK vs MRCK)?
Line 326-558. From point 3.4 to 3.11 onwards, it seems to be a different story and more consistent with a more straightforward storyline.
The manuscript lacks information about how B24 influences the structure of Leymus chinensis microbial communities.
Author Response
Dear Editors and Reviewers,
On behalf of my co-authors, we thank you very much for giving us an opportunity to revise our manuscript. We thank the reviewers for the thoughtful and valuable comments and suggestions on the manuscript. We have studied the comments carefully and have made revision accordingly (The manuscript is marked in red font). In addition, we discovered some other problems in the manuscript, which we have also revised. The major changes are described in detail in the attached point-by-point answers for your further evaluation. The comments are listed below in black font, and our responses are provided in blue font.
We would like to express our great appreciation to you and reviewers for comments on our paper. Looking forward to hearing from you. Thank you and best regards.
Sincerely,
Weibo Ren
School of Ecology and Environment
Inner Mongolia University
On behalf of all authors.
Reviewer 2
The manuscript presents a study focused on two main aspects: a) the bacterial diversity associated with Leymus chinensis under no grazing and overgrazing conditions, and b) the effects of Paraburkholderia graminis B24 on the transcriptomic and metabolomic profiles of Leymus chinensis under mown and unmown conditions.
However, the overall storyline is unclear and insufficiently developed. It appears that the authors are attempting to combine two distinct studies within a single manuscript, but there is a lack of coherence and integration between them. This disconnection—or possibly a lack of background information—hinders the reader's ability to grasp the aims and implications of the study entirely.
Although the manuscript provides potentially valuable descriptive insights at the transcriptomic and metabolomic levels, the methodology section lacks organization and appears incomplete, which makes the experimental workflow difficult to follow.
The topic is interesting, and the manuscript has potential for publication, but major revisions are required to improve the study's clarity, coherence, and completeness.
General comments
Comments 1: Please adjust the manuscript to the requirements of the Microorganism MDPI journal. According to the journal's requirements, it lacks information on the corresponding author and email address.
Response 1: Thank you for your comments. We have added the corresponding author and email address on Page 1, Line 9.
Comments 2: Please write in italics the scientific names of microorganisms, plants, and genes.
Response 2: Thank you for your valuable suggestion. According to your comment, we have carefully revised the manuscript to ensure that all scientific names of microorganisms , plants, and genes are now consistently formatted in italics throughout the text. These modifications have been highlighted in the revised manuscript.
Comments 3: Throughout the article, separate the text from the square brackets in the references. Although only a few examples are mentioned, this should be corrected throughout the article.
Response 3: Thank you for your careful review. We have revised the manuscript to ensure that all in-text citations are now properly separated from the preceding text with a space before the square brackets.
Changes
Comments 4: In Line 71. Please change "lay" for "play"
Response 4: Thank you for pointing out our mistake. We have changed "lay" for "play" on Page 2, Line 73.
Comments 5: Lines 81 and 82. Scientific names must be written in italics; please review them in all text of the manuscript.
Response 5: Thank you for your valuable suggestion. We have thoroughly reviewed the entire manuscript and ensured that all scientific names are correctly italicized throughout the text.
Comments 6: Line 93. Azotobacter genus in italics and capital letters. Please review scientific names in the manuscript.
Response 6: Thank you for your comment. We have checked the manuscript carefully and made sure that all scientific names are written in italics and follow taxonomic conventions. For genus names like Azotobacter, we used italics with the first letter capitalized on Page 3, Line 97.
Comments 7: Line 96. Please separate "stress" from the square bracket
Response 7: Thank you for pointing out our mistake. We have separated "stress" from the square bracket on Page 3, Line 99.
Comments 8: Line 106: Please delete "remains"
Response 8: Thank you for pointing out our mistake. We have deleted "remains"
on Page 3, Line 110.
Comments 9: Lines 145-153. This section lacks coherence with the information presented in Section 2.1. In section 2.2, the isolation and characterization of Paraburkholderia graminis B24 is introduced directly, without proper context. Please provide additional information to clarify why isolate B24 was selected and why the focus was explicitly placed on Paraburkholderia.
Response 9: Thank you for your valuable suggestion. We have added a description to clarify that Paraburkholderia graminis, as a key bacterium enriched under overgrazing stress, may play an important role in alleviating grazing-induced stress. In addition, we revised “B24” to “strains” and clarified that strain B24 was identified as Paraburkholderia graminis based on 16S rDNA sequencing. These changes were made on Page 4, Lines 150-152 and Line 154.
Comments 10: Line 158-160: For reagents, please include CAS number, company, or reagent brand.
Response 10: Thank you for your helpful suggestion. We have added the CAS numbers and supplier information (company or reagent brand) on Page 4, Lines 165-169 .
Comments 11: Line 185. Please separate "stress" from the square bracket
Response 11: Thank you for pointing out our mistake. We have separated "stress" from the square bracket.
Comments 12: Line 191. Please express the colony-forming units per milliliter (CFU/mL) in exponential notation.
Response 12: Thank you for your helpful suggestion. We have revised the colony-forming units per milliliter on Page 5, Line 201.
Comments 13: Line 205-214. Where are the results that describe the influence of Paraburkholderia graminis B24 on the community structure of Leymus chinensis?
Response 13: Thank you for your helpful suggestion. We have added a description of the methods used for analyzing the community structure and indicator species in the rhizosphere soil of Leymus chinensis (Page 5, Lines 226-229).
Comments 14: Line 231-240. It should be specified that the exudates analyzed were those released by the plants collected in section 2.1. Additionally, the distinction should be made between these exudates and the metabolites produced in the Leymus chinensis roots due to the effect of Paraburkholderia graminis B24.
Response 14: Thank you for your valuable suggestion. We have added relevant descriptions to distinguish between the root exudates of L. chinensis collected after overgrazing treatment and the root metabolites of L. chinensis after inoculation with B24 (see Page 6, Lines 248-250). Furthermore, we specified that the exudates analyzed were those released by the plants collected in section 2.1 on Page 6, Lines 255-256.
Comments 15: Line 249-266. This lacks consistency or coherence with the culturable bacteria described in lines 267–272.
Response 15: Thank you for your insightful comment. We have revised accordingly on Page 6, Lines 283-287.
Comments 16: Line 272-280. Please specify whether the information shown in Figure 1 corresponds to 16S rRNA sequencing culturable rhizosphere bacteria from Leymus chinensis or is based on metagenomic analysis. Alternatively, clarify how this data was obtained.
Response 16: Thank you for your comment. Lines 272–280 describe the effects of long-term overgrazing on rhizosphere bacterial communities, with the corresponding methodological details provided in Sections 2.1 and 2.6. This part does not involve 16S rRNA sequencing but rather focuses on culturable rhizosphere bacteria isolated from L. chinensis.
Comments 17: Line 281-289. In the same context, is section 3.2 connected to section 3.1? or are they addressing different topics? Or does it refer to culturable microorganisms?
Response 17: Thank you for your comment. In Section 3.1, we confirmed that Paraburkholderia graminis were only present in the OG treatments and as a key bacterium may play an important role in alleviating grazing-induced stress. In Section 3.2, among the culturable bacteria we isolated, strain B24 was identified as Paraburkholderia graminis. Therefore, these two sections have an inevitable connection, but the described contents are inconsistent.
Comments 18: Line 269 and 296. Some parts of the text are redundant.
Response 18: Thank you for your insightful comment. Some redundant content has been deleted on Page 7, Line 285 and Page 8, Lines 313-314.
Comments 19: Line 290-302. Are these results related to the data presented in section 2.1? (line 134-144)
Response 18: Thank you for your comment. Yes, the results in Line 290-302 related to the data presented in section 2.1? (line 134-144). To express this point more clearly, we have added relevant descriptions on Page 8, Lines 309-311.
Comments 20: Line 293, 294, and 297. Is referring to Figures 1A, 1B, 1C, and 1D correct, or did you mean Figures 2A, 2B, 2C, and 2D?
Response 20: Thank you for pointing out our mistake. We have changed Fig. 1A, 1B, 1C, and 1D to Fig. 2A, 2B, 2C, and 2D on Page 8, Lines 311-320.
Comments 21: Line 303-318. How did they obtain the correlation? What are they analyzing? Are they culturable or non-culturable bacteria?
Response 21: Thank you for your comment. Based on our above research content, we identified key root exudates and differential bacterial genera. Therefore, we further analyzed the correlations between key root exudates and significantly enriched bacterial genera , with the methods for the correlation analysis described in lines 262–263. These bacteria were obtained through amplicon sequencing, and only a subset are culturable, such as Paraburkholderia graminis.
Comments 22: Figure 2A. Is it correct to say Down (72 genes) and Up (72 genes), or perhaps does it refer to differentially synthesized metabolites?
Response 22: Thank you for pointing out our mistake. We are sorry for our negligence. We have changed“genes”to“DEMs”on Figure 2A.
Comments 23: Line 326-329. Are these compounds the most commonly synthesized in overgrazing? Or why were they chosen?
Response 23: Thank you for your insightful comment. These compounds were significantly enriched under overgrazing, and based on the results of the correlation analysis, we believe that they play a key role in promoting bacterial chemotaxis and biofilm formation. We have added relevant descriptions to address this point (see Page 10, Lines 347-351).
Comments 24: Line 389-390. What is the meaning of (RCK vs R24, MRCK vs MR24, and RCK vs MRCK)?
Response 24: Thank you for your comment. We have added relevant descriptions on the meaning of (RCK vs R24, MRCK vs MR24, and RCK vs MRCK) on Page 12, Lines 405-408.
Comments 25: Line 326-558. From point 3.4 to 3.11 onwards, it seems to be a different story and more consistent with a more straightforward storyline.
Response 25: Thank you for the valuable comment. I will further summarize the overall content of my research to help you better understand the main storyline. Our study first analyzed the impact of overgrazing on the rhizosphere bacterial community of L. chinensis and identified a key bacterium, Paraburkholderia graminis, which was present only under overgrazing conditions (Section 3.1). The bacterium was then isolated and purified (Section 3.2), and designated as strain B24. Subsequently, the interactions between B24 and key root exudates were investigated (Sections 3.3 and 3.4). A pot experiment was conducted with B24 inoculation. By analyzing plant growth traits, physiological parameters, and integrating transcriptomic and metabolomic analyses of L. chinensis, the effect of B24 on plant regeneration after mowing was explored (Sections 3.5–3.11). Ultimately, the study clarified the synergistic regulatory mechanisms by which PGPR (B24) help plants cope with the overgrazing stress by regulating specific root exudates, including amino acids, alkaloids, and organic acids.
Comments 26: The manuscript lacks information about how B24 influences the structure of Leymus chinensis microbial communities.
Response 26: Thank you for the valuable comment. We acknowledge the importance of understanding how strain B24 affects the microbial community structure of Leymus chinensis. However, this study primarily focused on the how L. chinensis responds to overgrazing stress by regulating its synergistic interactions with PGPR through root exudates, thereby alleviating grazing-induced stress. Detailed microbial community responses to B24 inoculation are beyond the scope of the current work but are part of our ongoing research. We have conducted preliminary research confirming that B24 inoculation indeed affects the composition and diversity of bacterial communities (see figure below). In the future, we will further investigate its impact on indicator species. We have conducted preliminary research confirming that B24 inoculation indeed affects the composition and diversity of bacterial communities (see figure below). In the future, we will further investigate its effects on indicator species, community structure, and function.
Figure 1 B24 inoculation on the composition and diversity of rhizosphere bacterial community in L. chinensis. (A) Phyla level bacterial community composition; (B) Genus level bacterial community composition; (C) ACE abundance index of bacterial community; (D) Simpson diversity index of bacterial community.

Round 2
Reviewer 2 Report
Comments and Suggestions for Authors
The manuscript has been significantly improved and is now nearing the standards required for publication in Microorganisms. However, several minor revisions are still necessary prior to acceptance. The authors should address specific formatting issues, such as ensuring that “OD470” (line 177) is properly written with the wavelength as a subscript (i.e., OD470nm). Additionally, reference citations should be clearly separated from the preceding text. For example, in line 134: “Ren et al. (2008)[28]” should be corrected to “Ren et al. (2008) [28]”.
Moreover, the manuscript would benefit from a thorough English language revision by a native speaker to improve clarity, grammar, and overall readability.
Although most results are supported by corresponding methodological descriptions, the sequence of the methods does not follow a logical or chronological structure consistent with the order of the results. This disrupts the narrative flow and leaves some ideas underdeveloped, which may hinder reader comprehension. For instance, Sections 2.3 (chemotaxis and biofilm) and 2.8 (metabolomics) appear later in the results, despite being introduced earlier in the methodology. It is therefore recommended to reorganize the methodology section to align with the experimental flow presented in the results—for example: begin with Sections 2.1 and 2.6 (sampling and 16S rDNA sequencing), followed by 2.2, 2.8, 2.3, 2.4, 2.5, and 2.7. A similar reordering of the results section is also advised to maintain consistency and enhance coherence throughout the manuscript.
As an example, it is important to clarify that the results presented in Section 3.1 were derived from 16S rDNA sequencing, which identified Paraburkholderia graminis B24 as the most abundant species. Therefore, it may be considered a potential biomarker for overgrazing. Based on these molecular findings, cultivable microorganisms were subsequently isolated. Paraburkholderia graminis B24 was again identified as the dominant species and was selected for further characterization, including its plant growth-promoting traits, effects on host plants, metabolomic profiling, and transcriptomic analysis.
Once these revisions—particularly the formatting, grammatical, and structural improvements—are addressed, I consider the manuscript suitable for publication.
Comments on the Quality of English LanguageThe manuscript should be thoroughly reviewed and edited by a native English speaker to ensure proper grammar, spelling, and clarity throughout the text.
Author Response
Dear Reviewer,
On behalf of my co-authors, we thank you very much for giving us an opportunity to revise our manuscript. We thank the reviewers for the thoughtful and valuable comments and suggestions on the manuscript. We have studied the comments carefully and have made revision accordingly (The manuscript is marked in red font). In addition, we discovered some other problems in the manuscript, which we have also revised. The major changes are described in detail in the attached point-by-point answers for your further evaluation. The comments are listed below in black font, and our responses are provided in blue font.
We would like to express our great appreciation to you and reviewers for comments on our paper. Looking forward to hearing from you. Thank you and best regards.
Sincerely,
Weibo Ren
School of Ecology and Environment
Inner Mongolia University
On behalf of all authors.
Reviewer 1
Comments 1: The manuscript has been significantly improved and is now nearing the standards required for publication in Microorganisms. However, several minor revisions are still necessary prior to acceptance. The authors should address specific formatting issues, such as ensuring that “OD470” (line 177) is properly written with the wavelength as a subscript (i.e., OD470nm). Additionally, reference citations should be clearly separated from the preceding text. For example, in line 134: “Ren et al. (2008)[28]” should be corrected to “Ren et al. (2008) [28]”.
Response 1: Thank you for your comments. We have made corresponding modifications in line 210 and 139.
Comments 2: Moreover, the manuscript would benefit from a thorough English language revision by a native speaker to improve clarity, grammar, and overall readability.
Response 2: Thank you for your valuable suggestion. We appreciate your comment regarding the language quality of the manuscript. We have carefully revised the entire manuscript to improve clarity, grammar, and readability.
Comments 3: Although most results are supported by corresponding methodological descriptions, the sequence of the methods does not follow a logical or chronological structure consistent with the order of the results. This disrupts the narrative flow and leaves some ideas underdeveloped, which may hinder reader comprehension. For instance, Sections 2.3 (chemotaxis and biofilm) and 2.8 (metabolomics) appear later in the results, despite being introduced earlier in the methodology. It is therefore recommended to reorganize the methodology section to align with the experimental flow presented in the results—for example: begin with Sections 2.1 and 2.6 (sampling and 16S rDNA sequencing), followed by 2.2, 2.8, 2.3, 2.4, 2.5, and 2.7. A similar reordering of the results section is also advised to maintain consistency and enhance coherence throughout the manuscript.
As an example, it is important to clarify that the results presented in Section 3.1 were derived from 16S rDNA sequencing, which identified Paraburkholderia graminis B24 as the most abundant species. Therefore, it may be considered a potential biomarker for overgrazing. Based on these molecular findings, cultivable microorganisms were subsequently isolated. Paraburkholderia graminis B24 was again identified as the dominant species and was selected for further characterization, including its plant growth-promoting traits, effects on host plants, metabolomic profiling, and transcriptomic analysis.
Response 3: Thank you very much for your thoughtful and constructive comment. We agree that the sequence of the Methods and Results sections should follow a logical and chronological structure to enhance clarity and coherence. In response, we have reorganized the Materials and Methods and Results section to better align with the experimental flow and the presentation of the Results.